# On the Robustness of Randomized Ensembles to Adversarial Perturbations

## Abstract

Randomized ensemble classifiers (RECs), where *one* classifier is randomly selected during inference, have emerged as an attractive alternative to traditional ensembling methods for realizing adversarially robust classifiers with limited compute requirements. However, recent works have shown that existing methods for constructing RECs are more vulnerable than initially claimed, casting major doubts on their efficacy and prompting *fundamental* questions such as: "*When are RECs useful?*", "*What are their limits?*", and "*How do we train them?*". In this work, we first demystify RECs as we derive fundamental results regarding their theoretical limits, necessary and sufficient conditions for them to be useful, and more. Leveraging this new understanding, we propose a new boosting algorithm (BARRE) for training robust RECs, and empirically demonstrate its effectiveness at defending against strong $\ell_\infty$ norm-bounded adversaries across various network architectures and datasets. *Our code is submitted as part of the supplementary material, and will be publicly released on GitHub.*

## 1 Introduction

Defending deep networks against adversarial perturbations (Szegedy et al., 2013; Biggio et al., 2013; Goodfellow et al., 2014) remains a difficult task. Several proposed defenses (Papernot et al., 2016; Pang et al., 2019; Yang et al., 2019; Sen et al., 2019; Pinot et al., 2020) have been subsequently "broken" by stronger adversaries (Carlini & Wagner, 2017; Athalye et al., 2018; Tramèr et al., 2020; Dbouk & Shanbhag, 2022), whereas strong defenses (Cisse et al., 2017; Tramèr et al., 2018; Cohen et al., 2019), such as adversarial training (AT) (Goodfellow et al., 2014; Zhang et al., 2019; Madry et al., 2018), achieve unsatisfactory levels of robustness[1].

A popular belief in the adversarial community is that single model defenses, e.g., AT, lack the capacity to defend against all possible perturbations, and that constructing an ensemble of diverse, often smaller, models should be more cost-effective (Pang et al., 2019; Kariyappa & Qureshi, 2019; Pinot et al., 2020; Yang et al., 2020b; 2021; Abernethy et al., 2021; Zhang et al., 2022). Indeed, recent *deterministic* robust ensemble methods, such as MRBoost (Zhang et al., 2022), have been successful at achieving higher robustness compared to AT baselines using the same network architecture, at the expense of $4\times$ more compute (see Fig. 1). In fact, Fig 1 indicates that one can simply adversarially training *larger* deep nets that can match the robustness and compute requirements of MRBoost models, rendering state-of-the-art boosting techniques obsolete for designing classifiers that are both *robust* and *efficient*.

In contrast, *randomized* ensembles, where one classifier is randomly selected during inference, offer a unique way of ensembling that can operate with limited compute resources. However, the recent work of Dbouk & Shanbhag (2022) has cast major concerns regarding their efficacy, as they successfully compromised the state-of-the-art randomized defense of Pinot et al. (2020) by large margins using their proposed ARC adversary. Furthermore, there is an apparent lack of *proper* theory on the robustness of randomized ensembles, as fundamental questions such as: "when does randomization help?" or "how to find the optimal sampling probability?" remain unanswered.

**Contributions**. In this work, we first provide a theoretical framework for analyzing the adversarial robustness of randomized ensmeble classifiers (RECs). Our theoretical results enable us to better

---

[1]when compared to the high clean accuracy achieved in a non-adversarial setting

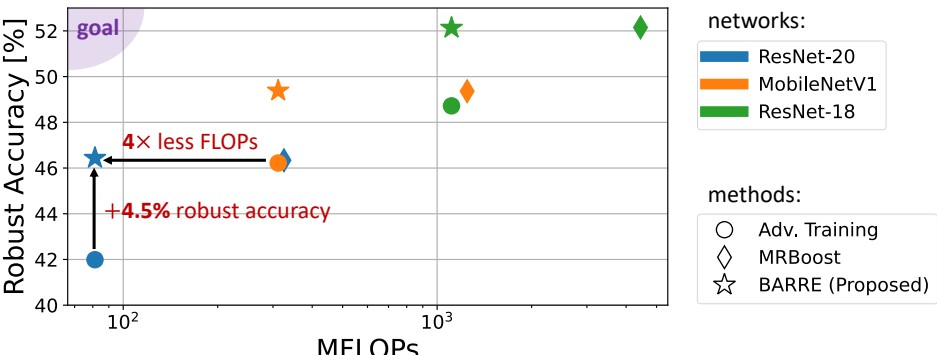

Figure 1: The efficacy of employing randomized ensembles (⋆) for achieving robust and efficient inference compared to AT (●) and deterministic ensembling MRBoost (♦) on CIFAR-10. Robustness is measured using the standard $\ell_\infty$ norm-bounded adversary with radius $\epsilon = 8/255$.

understand randomized ensembles, revealing interesting and useful answers regarding their limits, necessary and sufficient conditions for them to be useful, and efficient methods for finding the optimal sampling probability. Next, guided by our threoretical results, we propose BARRE, a new boosting algorithm for training robust randomized ensemble classifiers achieving state-of-the-art robustness. We validate the effectiveness of BARRE via comprehensive experiments across multiple network architectures and datasets, thereby demonstrating that RECs can achieve similar robustness to AT and MRBoost, at a *fraction* of the computational cost (see Fig. 1).

## 2 BACKGROUND AND RELATED WORK

**Adversarial Robustness**. Deep neural networks are known to be vulnerable to adversarial perturbations (Szegedy et al., 2013; Biggio et al., 2013). In an attempt to robustify deep nets, several defense methods have been proposed (Katz et al., 2017; Madry et al., 2018; Cisse et al., 2017; Zhang et al., 2019; Yang et al., 2020b; Zhang et al., 2022; Tjeng et al., 2018; Xiao et al., 2018; Raghunathan et al., 2018; Yang et al., 2020a). While some heuristic-based *empirical* defenses have later been broken by better adversaries (Carlini & Wagner, 2017; Athalye et al., 2018; Tramèr et al., 2020), strong defenses, such as adversarial training (AT) (Goodfellow et al., 2014; Madry et al., 2018; Zhang et al., 2019), remain unbroken but achieve unsatisfactory levels of robustness.

**Ensemble Defenses**. Building on the massive success of classic ensemble methods in machine learning (Breiman, 1996; Freund & Schapire, 1997; Dietterich, 2000), robust ensemble methods (Kariyappa & Qureshi, 2019; Pang et al., 2019; Sen et al., 2019; Yang et al., 2020b; 2021; Abernethy et al., 2021; Zhang et al., 2022) have emerged as a natural solution to compensate for the unsatisfactory performance of existing single-model defenses, such as AT. Earlier works (Kariyappa & Qureshi, 2019; Pang et al., 2019; Sen et al., 2019) relied on heuristic-based techniques for inducing diversity within the ensembles, and have been subsequently shown to be weak (Tramèr et al., 2020; Athalye et al., 2018). Recent methods, such as RobBoost (Abernethy et al., 2021) and MRBoost (Zhang et al., 2022), formulate the design of robust ensembles from a margin boosting perspective, achieving state-of-the-art robustness for deterministic ensemble methods. This achievement comes at a massive $(4 - 5\times)$ increase in compute requirements, as each inference requires executing all members of the ensemble, deeming them unsuitable for safety-critical edge applications (Guo et al., 2020; Sehwag et al., 2020; Dbouk & Shanbhag, 2021). Randomized ensembles (Pinot et al., 2020), where one classifier is chosen randomly during inference, offer a more compute-efficient alternative. However, their ability to defend against strong adversaries remains unclear (Dbouk & Shanbhag, 2022; Zhang et al., 2022). In this work, we show that randomized ensemble classifiers can be effective at defending against adversarial perturbations, and propose a boosting algorithm for training such ensembles, thereby achieving high levels of robustness with limited compute requirements.

**Randomized Defenses**. A randomized defense, where the defender adopts a random strategy for classification, is intuitive: if the defender does not know what is the exact policy used for a certain input, then one expects that the adversary will struggle *on average* to fool such a defense. Theoretically, Bayesian Neural Nets (BNNs) (Neal, 2012) have been shown to be robust (in the large data limit) to gradient-based attacks (Carbone et al., 2020), whereas Pinot et al. (2020) has shown that a randomized ensemble classifier (REC) with higher robustness exists for every deterministic classifier. However, realizing strong and practical randomized defenses remains elusive as BNNs are too computationally prohibitive and existing methods (Xie et al., 2018; Dhillon et al., 2018; Yang et al., 2019) often end up being compromised by adaptive attacks (Athalye et al., 2018; Tramèr et al., 2020). Even BAT, the proposed method of Pinot et al. (2020) for robust RECs, was recently broken by Zhang et al. (2022); Dbouk & Shanbhag (2022). In contrast, our work first demystifies randomized ensembles as we derive fundamental results regarding the limit of RECs, necessary and sufficient conditions for them to be useful, and efficient methods for finding the optimal sampling probability. Empirically, our proposed boosting algorithm (BARRE) can successfully train robust RECs, achieving state-of-the-art robustness for RECs.

## 3    PRELIMINARIES & PROBLEM SETUP

**Notation**. Let $\mathcal{F} = \{f_1, ..., f_M\}$ be a collection of $M$ arbitrary $C$-ary classifiers $f_i : \mathbb{R}^d \to [C]$. A soft classifier, denoted by $\tilde{f} : \mathbb{R}^d \to \mathbb{R}^C$, can be used to construct a hard classifier $f(\mathbf{x}) = \arg\max_{c\in[C]}[\tilde{f}(\mathbf{x})]_c$, where $[\mathbf{v}]_c = v_c$. We use the notation $f(\cdot|\boldsymbol{\theta})$ to represent *parametric* classifiers where $f$ is a *fixed* mapping and $\boldsymbol{\theta} \in \Theta$ represents the learnable parameters. Let $\Delta_M = \{\mathbf{v} \in [0,1]^M : \sum v_i = 1\}$ be the probability simplex of dimension $M-1$. Given a probability vector $\boldsymbol{\alpha} \in \Delta_M$, we construct a randomized ensemble classifier (REC) $f_{\boldsymbol{\alpha}}$ such that $f_{\boldsymbol{\alpha}}(\mathbf{x}) = f_i(\mathbf{x})$ with probability $\alpha_i$. In contrast, traditional ensembling methods construct a deterministic ensemble classifier (DEC) using the soft classifiers as follows[2]: $\bar{f}(\mathbf{x}) = \arg\max_{c\in[C]}[\sum_{i=1}^M \tilde{f}_i(\mathbf{x})]_c$. Denote $\mathbf{z} = (\mathbf{x}, y) \in \mathbb{R}^d \times [C]$ as a feature-label pair that follows some unknown distribution $\mathcal{D}$. Let $\mathcal{S} \subset \mathbb{R}^d$ be a closed and bounded set representing the attacker's perturbation set. A typical choice of $\mathcal{S}$ in the adversarial community is the $\ell_p$ ball of radius $\epsilon$: $\mathcal{B}_p(\epsilon) = \{\boldsymbol{\delta} \in \mathbb{R}^d : \|\boldsymbol{\delta}\|_p \le \epsilon\}$. For a classifier $f_i \in \mathcal{F}$ and data-point $\mathbf{z} = (\mathbf{x}, y)$, define $\mathcal{S}_i(\mathbf{z}) = \{\boldsymbol{\delta} \in \mathcal{S} : f_i(\mathbf{x}+\boldsymbol{\delta}) \ne y\}$ to be the set of valid *adversarial* perturbations to $f_i$ at $\mathbf{z}$.

**Definition 1.** For any (potentially random) classifier $f$, define the *adversarial risk* $\eta$:

$$\eta(f) = \mathbb{E}_{\mathbf{z}\sim\mathcal{D}}\left[\max_{\boldsymbol{\delta}\in\mathcal{S}}\mathbb{E}_f\left[\mathbb{1}\left\{f(\mathbf{x}+\boldsymbol{\delta}) \ne y\right\}\right]\right] \tag{1}$$

The adversarial risk measures the robustness of $f$ on average in the presence of an adversary (attacker) restricted to the set $\mathcal{S}$. For the special case of $\mathcal{S} = \{\mathbf{0}\}$, the adversarial risk reduces to the *standard risk* of $f$:

$$\eta_0(f) = \mathbb{E}_{\mathbf{z}\sim\mathcal{D}}\left[\mathbb{E}_f\left[\mathbb{1}\left\{f(\mathbf{x}) \ne y\right\}\right]\right] = \mathbb{P}\left\{f(\mathbf{x}) \ne y\right\} \tag{2}$$

The more commonly reported robust accuracy of $f$, i.e., accuracy against adversarially perturbed inputs, can be directly computed from $\eta(f)$. The same can be said for the clean accuracy and $\eta_0(f)$.

When working with an REC $f_{\boldsymbol{\alpha}}$, the adversarial risk can be expressed as:

$$\eta(f_{\boldsymbol{\alpha}}) \equiv \eta(\boldsymbol{\alpha}) = \mathbb{E}_{\mathbf{z}\sim\mathcal{D}}\left[\max_{\boldsymbol{\delta}\in\mathcal{S}}\sum_{i=1}^M \alpha_i \mathbb{1}\left\{f_i(\mathbf{x}+\boldsymbol{\delta}) \ne y\right\}\right] \tag{3}$$

where we use the notation $\eta(\boldsymbol{\alpha})$ whenever the collection $\mathcal{F}$ is fixed. Let $\{\mathbf{e}_i\}_{i=1}^M \subset \{0,1\}^M$ be the standard basis vectors of $\mathbb{R}^M$, then we employ the notation $\eta(f_i) = \eta(f_{\mathbf{e}_i}) \equiv \eta(\mathbf{e}_i) = \eta_i$.

## 4    THE ADVERSARIAL RISK OF A RANDOMIZED ENSEMBLE CLASSIFIER

In this section, we develop our main theoretical findings regarding the adversarial robustness of any randomized ensemble classifier. Detailed proofs of all statements and theorems can be found in Appendix B.

---

[2] the normalizing constant $\frac{1}{M}$ does not affect the classifier output

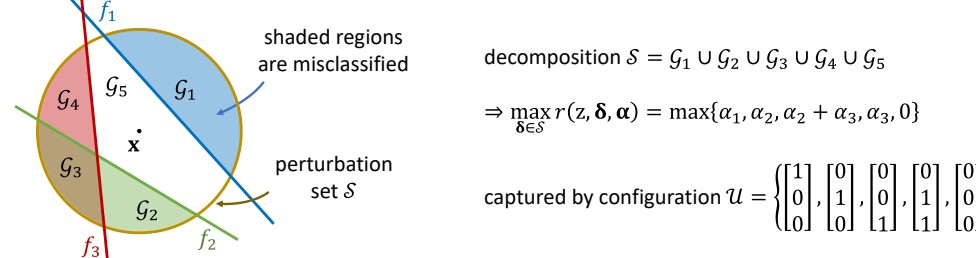

Figure 2: Illustration of the equivalence in (6) using an example of three classifiers in $\mathbb{R}^2$. The shaded areas represent regions in the attacker-restricted input space where each classifier makes an error. All classifiers correctly classify $\mathbf{x}$. The set $\mathcal{U}$ uniquely captures the interaction between $\mathbf{z}$ and $f_1$, $f_2$, & $f_3$ inside $\mathcal{S}$.

### 4.1 PROPERTIES OF $\eta$

We start with the following statement:

**Proposition 1.** *For any $\mathcal{F} = \{f_i\}_{i=1}^M$, perturbation set $\mathcal{S} \subset \mathbb{R}^d$, and data distribution $\mathcal{D}$, the adversarial risk $\eta$ is a piece-wise linear convex function $\forall \boldsymbol{\alpha} \in \Delta_M$. Specifically, $\exists K \in \mathbb{N}$ configurations $\mathcal{U}_k \subseteq \{0,1\}^M \; \forall k \in [K]$ and p.m.f. $\mathbf{p} \in \Delta_K$ such that:*

$$\eta(\boldsymbol{\alpha}) = \sum_{k=1}^K \left( p_k \cdot \max_{\mathbf{u} \in \mathcal{U}_k} \left\{ \mathbf{u}^\top \boldsymbol{\alpha} \right\} \right) \tag{4}$$

Before we explain the intuition behind Proposition 1, we first make the following observations:

**Generality**. Proposition 1 makes no assumptions about the classifiers $\mathcal{F}$, i.e., it applies even to the enigmatic deep nets. While the majority of theoretical results in the literature have been restricted to $\ell_p$-bounded adversaries, Proposition 1 holds for any closed and bounded perturbation set $\mathcal{S}$. This is crucial, as real-world attacks are often not restricted to $\ell_p$ balls around the input (Liu et al., 2018; Duan et al., 2020). This generality is further inherited by all of our results, as they build on Proposition 1.

**Analytic Form**. Proposition 1 allows us to re-write the adversarial risk in (3) using the analytic form in (4), which is much simpler to analyze and work with. In fact, the analytic form in (4) enables us to derive our main theoretical results in Sections 4.2 & 4.3, which include *tight fundamental bounds* on $\eta$.

**Optimal Sampling**. The convexity of $\eta$ implies that any local minimum $\boldsymbol{\alpha}^*$ is also a global minimum. The probability simplex is a closed convex set, thus a global minimum, which need not be unique, is always achievable. Since $\eta$ is piece-wise linear, then there always exists a *finite* set of candidate solutions for $\boldsymbol{\alpha}^*$. For $M \leq 3$, we efficiently enumerate all candidates in Section 4.2, eliminating the need for any sophisticated search method. For larger $M$ however, enumeration becomes intractable. In Section 4.4, we construct an optimal algorithm for finding $\boldsymbol{\alpha}^*$ by leveraging the classic sub-gradient method (Shor, 2012) for optimizing sub-differentiable functions.

**Intuition**. Consider a data-point $\mathbf{z} \in \mathbb{R}^d \times [C]$, then for any $\boldsymbol{\delta} \in \mathcal{S}$ and $\boldsymbol{\alpha} \in \Delta_M$ we have the per sample risk:

$$r(\mathbf{z}, \boldsymbol{\delta}, \boldsymbol{\alpha}) = \sum_{i=1}^M \alpha_i \mathbb{1}\{f_i(\mathbf{x} + \boldsymbol{\delta}) \neq y\} = \mathbf{u}^\top \boldsymbol{\alpha} \tag{5}$$

where $\mathbf{u} \in \{0,1\}^M$ such that $u_i = 1$ if and only if $\boldsymbol{\delta}$ is adversarial to $f_i$ at $\mathbf{z}$. Since $\mathbf{u}$ is independent of $\boldsymbol{\alpha}$, we thus obtain a many-to-one mapping from $\boldsymbol{\delta} \in \mathcal{S}$ to $\mathbf{u} \in \{0,1\}^M$. Therefore, for any $\boldsymbol{\alpha}$ and $\mathbf{z}$, we can always decompose the perturbation set $\mathcal{S}$, i.e., $\mathcal{S} = \mathcal{G}_1 \cup ... \cup \mathcal{G}_n$, into $n \leq 2^M$ subsets, such that: $\forall \boldsymbol{\delta} \in \mathcal{G}_j : r(\mathbf{z}, \boldsymbol{\delta}, \boldsymbol{\alpha}) = \boldsymbol{\alpha}^\top \mathbf{u}_j$ for some binary vector $\mathbf{u}_j$ independent of $\boldsymbol{\alpha}$. Let $\mathcal{U} = \{\mathbf{u}_j\}_{j=1}^n$ be the collection of these vectors, then we can write:

$$\max_{\boldsymbol{\delta} \in \mathcal{S}} r(\mathbf{z}, \boldsymbol{\delta}, \boldsymbol{\alpha}) = \max_{\boldsymbol{\delta} \in \mathcal{G}_1 \cup ... \cup \mathcal{G}_n} r(\mathbf{z}, \boldsymbol{\delta}, \boldsymbol{\alpha}) = \max_{j \in [n]} \left\{ \max_{\boldsymbol{\delta} \in \mathcal{G}_j} r(\mathbf{z}, \boldsymbol{\delta}, \boldsymbol{\alpha}) \right\} = \max_{\mathbf{u} \in \mathcal{U}} \left\{ \mathbf{u}^\top \boldsymbol{\alpha} \right\} \tag{6}$$

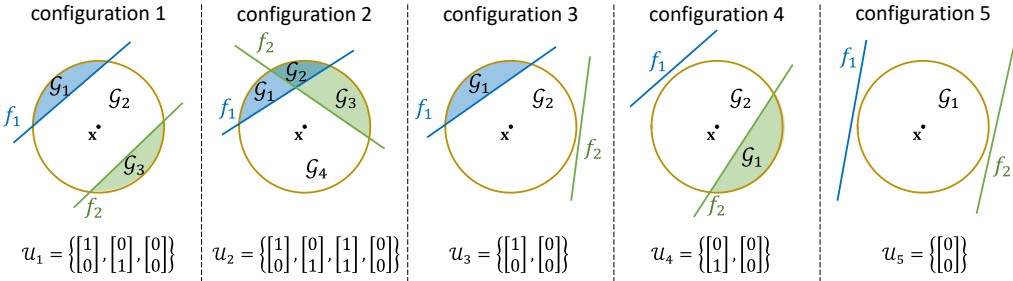

Figure 3: Enumeration of all $K = 5$ unique configurations with two classifiers and a data-point around a set $\mathcal{S}$. Note that since $\alpha_i \geq 0 \; \forall i$, the $\mathbf{0}$ vector is redundant in $\mathcal{U}_k$ for $k \in [4]$, which explains why $K = 5$ and not more.

The main idea behind the equivalence in (6) is that we can represent any configuration of classifiers, data-point and perturbation set using a unique set of binary vectors $\mathcal{U}$. For example, Fig. 2 pictorially depicts this equivalence using a case of $M = 3$ classifiers in $\mathbb{R}^2$ with $\mathcal{S} = \mathcal{B}_2(\epsilon)$. This equivalence is the key behind Proposition 1, since the point-wise max term in (6) is piece-wise linear and convex $\forall \boldsymbol{\alpha} \in \Delta_M$. Finally, Proposition 1 holds due to the pigeon-hole principle and the linearity of expectation.

## 4.2 SPECIAL CASE OF TWO CLASSIFIERS

With two classifiers only, we can leverage the analytic form of $\eta$ in (4) and enumerate all possible classifiers/data-point configurations around $\mathcal{S}$ by enumerating all configurations $\mathcal{U}_k \subseteq \{0, 1\}^2$. Specifically, Fig. 3 visualizes all $K = 5$ such unique configurations, which allows us to write $\forall \boldsymbol{\alpha} \in \Delta_2$:

$$\eta(\boldsymbol{\alpha}) = p_1 \cdot \max\{\alpha_1, \alpha_2\} + p_2 \cdot 1 + p_3 \cdot \alpha_1 + p_4 \cdot \alpha_2 + p_5 \cdot 0 \tag{7}$$

where $\mathbf{p} \in \Delta_5$ is the p.m.f. of "binning" any data-point $\mathbf{z}$ into any of the five configurations, under the data distribution $\mathbf{z} \sim \mathcal{D}$. Using (7), we obtain the following result:

**Theorem 1.** *For any two classifiers $f_1$ and $f_2$ with individual adversarial risks $\eta_1$ and $\eta_2$, respectively, subject to a perturbation set $\mathcal{S} \subset \mathbb{R}^d$ and data distribution $\mathcal{D}$, if:*

$$\mathbb{P}\{\mathbf{z} \in \mathcal{R}_1\} > |\eta_1 - \eta_2| \tag{8}$$

*where:*

$$\mathcal{R}_1 = \{\mathbf{z} \in \mathbb{R}^d \times [C] : \mathcal{S}_1(\mathbf{z}) \neq \varnothing, \mathcal{S}_2(\mathbf{z}) \neq \varnothing, \mathcal{S}_1(\mathbf{z}) \cap \mathcal{S}_2(\mathbf{z}) = \varnothing\} \tag{9}$$

*then the optimal sampling probability $\boldsymbol{\alpha}^* = [1/2 \; 1/2]^\top$ uniquely minimizes $\eta(\boldsymbol{\alpha})$ resulting in $\eta(\boldsymbol{\alpha}^*) = \frac{1}{2}(\eta_1 + \eta_2 - \mathbb{P}\{\mathbf{z} \in \mathcal{R}_1\})$. Otherwise, $\boldsymbol{\alpha}^* \in \{\mathbf{e}_1, \mathbf{e}_2\}$ minimizes $\eta(\boldsymbol{\alpha})$, where $\mathbf{e}_i$s are the standard basis vectors of $\mathbb{R}^2$.*

Theorem 1 provides us with a *complete* description of how randomized ensembles operate when $M = 2$. We discuss its implications below:

**Interpretation**. Theorem 1 states that randomization is *guaranteed* to help when the condition in (8) is satisfied, i.e., when the probability of data-points $\mathbf{z}$ ($\mathbb{P}\{\mathbf{z} \in \mathcal{R}_1\}$) for which it is possible to find adversarial perturbations that can fool $f_1$ or $f_2$ but not both (see configuration 1 in Fig. 3), is greater than the absolute difference ($|\eta_1 - \eta_2|$) of the individual classifiers' adversarial risks. Consequently, if the adversarial risks of the classifiers are heavily skewed, i.e., $|\eta_1 - \eta_2|$ is large, then randomization is less likely to help, since condition (8) becomes harder to satisfy. This, in fact, is the case for BAT defense (Pinot et al., 2020) since it generates two classifiers with $\eta_1 < 1$ and $\eta_2 = 1$. Theorem 1 indicates that adversarial defenses should strive to achieve $\eta_1 \approx \eta_2$ for randomization to be effective. In practice, it is very difficult to make $\mathbb{P}\{\mathbf{z} \in \mathcal{R}_1\}$ very large compared to $\eta_1$ and $\eta_2$ due to transferability of adversarial perturbations.

**Optimality Condition**. In fact, the condition in (8) is actually a *necessary* and *sufficient* condition for $\eta(\boldsymbol{\alpha}^*) < \min\{\eta_1, \eta_2\}$. That is, a randomized ensemble of $f_1$ and $f_2$ is *guaranteed* to achieve smaller adversarial risk than either $f_1$ of $f_2$ if and only if (8) holds. This also implies that it is

*impossible* to have a nontrivial[3] unique global minimizer other than $\boldsymbol{\alpha}^* = [^1/_2 \ ^1/_2]^\top$, which provides further theoretical justification for why the BAT defense in (Pinot et al., 2020) does not work, where $\boldsymbol{\alpha}^* = [0.9 \ 0.1]^\top$ was claimed to be a unique optimum (obtained via sweeping $\boldsymbol{\alpha}$).

**Simplified Search**. Theorem 1 *eliminates* the need for sweeping $\boldsymbol{\alpha}$ to find the optimal sampling probability $\boldsymbol{\alpha}^*$ when working with $M = 2$ classifiers as done in (Pinot et al., 2020; Dbouk & Shanbhag, 2022). We only need to evaluate $\eta\left([^1/_2 \ ^1/_2]^\top\right)$ and check if it is smaller than $\min\{\eta_1, \eta_2\}$ to choose our optimal sampling probability. In Appendix C.1, we extend this result for $M = 3$. Interestingly, Vorobeychik & Li (2014) derive a similar result for $M = 2$ for a different problem of an adversary attempting to reverse engineer the defender's classifier via queries.

**Theoretical Limit**. From Theorem 1, we can directly obtain a *tight* bound on the adversarial risk:

**Corollary 1.** *For any two classifiers $f_1$ and $f_2$ with individual adversarial risks $\eta_1$ and $\eta_2$, respectively, perturbation set $\mathcal{S}$, and data distribution $\mathcal{D}$:*

$$\min_{\boldsymbol{\alpha} \in \Delta_2} \eta(\boldsymbol{\alpha}) = \eta(\boldsymbol{\alpha}^*) \geq \min\left\{\frac{1}{2}\max\{\eta_1, \eta_2\}, \min\{\eta_1, \eta_2\}\right\}. \tag{10}$$

In other words, it is impossible for a REC with $M = 2$ classifiers to achieve a risk smaller than the RHS in (10). In the next section, we derive a more general version of this bound for arbitrary $M$.

## 4.3 TIGHT FUNDAMENTAL BOUNDS

A fundamental question remains to be answered: given an ensemble $\mathcal{F}$ of $M$ classifiers with adversarial risks $\eta_1, ..., \eta_M$, what is the tightest bound we can provide for the adversarial risk $\eta(\boldsymbol{\alpha})$ of a randomized ensemble constructed from $\mathcal{F}$? The following theorem answers this question:

**Theorem 2.** *For a perturbation set $\mathcal{S}$, data distribution $\mathcal{D}$, and collection of $M$ classifiers $\mathcal{F}$ with individual adversarial risks $\eta_i$ ($i \in [M]$) such that $0 < \eta_1 \leq ... \leq \eta_M \leq 1$, we have $\forall \boldsymbol{\alpha} \in \Delta_M$:*

$$\min_{k \in [M]}\left\{\frac{\eta_k}{k}\right\} \leq \eta(\boldsymbol{\alpha}) \leq \eta_M \tag{11}$$

*Both bounds are tight in the sense that if all that is known about the setup $\mathcal{F}$, $\mathcal{D}$, and $\mathcal{S}$ is $\{\eta_i\}_{i=1}^M$, then there exist no tighter bounds. Furthermore, the upper bound is always met if $\boldsymbol{\alpha} = \mathbf{e}_M$, and the lower bound (if achievable) can be met if $\boldsymbol{\alpha} = \left[\frac{1}{m} \ ... \ \frac{1}{m} \ 0 \ ... \ 0\right]^\top$, where $m = \arg\min_{k \in [M]}\{\frac{\eta_k}{k}\}$.*

**Upper bound**: The upper bound in (11) holds due to the convexity of $\eta$ (Proposition 1) and the fact $\Delta_M = \mathcal{H}\left(\{\mathbf{e}_i\}_{i=1}^M\right)$, where $\mathcal{H}(\mathcal{X})$ is the convex hull of the set of points $\mathcal{X}$.

**Implications of upper bound**: Intuitively, we expect that a randomized ensemble cannot be worse than the worst performing member (in this case $f_M$). A direct implication of this is that if all the members have similar robustness $\eta_i \approx \eta_j \ \forall i, j$, then randomized ensembling is *guaranteed* to either improve or achieve the same robustness. In contrast, deterministic ensemble methods that average logits (Zhang et al., 2022; Abernethy et al., 2021; Kariyappa & Qureshi, 2019) do not even satisfy this upper bound (see Appendix C.2). In other words, there are *no* worst-case performance guarantees with deterministic ensembling, even if all the classifiers are robust.

**Lower bound**: The main idea behind the proof of the lower bound in (11) is to show that $\forall \boldsymbol{\alpha} \in \Delta_M$:

$$\eta(\boldsymbol{\alpha}) \geq \sum_{i=1}^M \left((\eta_i - \eta_{i-1}) \cdot \max_{j \in \{i,...,M\}}\{\alpha_j\}\right) = h(\boldsymbol{\alpha}) \geq \min_{\boldsymbol{\alpha} \in \Delta_M} h(\boldsymbol{\alpha}) = h(\boldsymbol{\alpha}^*) = \frac{\eta_m}{m} \tag{12}$$

where $\eta_0 \doteq 0$, $m = \arg\min_{k \in [M]}\{\eta_k/k\}$, and $h$ can be interpreted as the adversarial risk of an REC constructed from an optimal set of classifiers $\mathcal{F}'$ with the same individual risks as $\mathcal{F}$. We make the following observations:

**Implications of lower bound**: The lower bound in (11) provides us with a *fundamental limit* on the adversarial risk of RECs viz., it is *impossible* for any REC constructed from $M$ classifiers with

---

[3]that is different than $\mathbf{e}_1$ or $\mathbf{e}_2$

sorted risks $\{\eta_i\}_{i=1}^M$ to achieve an adversarial risk smaller than $\min_{k \in [M]}\{\eta_k/k\} = \eta_m/m$. This limit is not always achievable and generalizes the one in (10) which holds for $M = 2$. Theorem 2 states that *if* the limit is achievable then the corresponding optimal sampling probability $\boldsymbol{\alpha}^* = \left[\frac{1}{m} \dots \frac{1}{m} 0 \dots 0\right]^\top$. Note that this does not imply that the optimal sampling probability is always equiprobable sampling $\forall \mathcal{F}$!

Additionally, the lower bound in (11) provides guidelines for robustifying individual classifiers in order for randomized ensembling to enhance the overall adversarial risk. Given classifiers $f_1, ..., f_m$ obtained via any sequential ensemble training algorithm, a good rule of thumb for the classifier obtained via the training iteration $m + 1$ is to have:

$$\eta_m \leq \eta_{m+1} \leq \left(1 + \frac{1}{m}\right)\eta_m \qquad (13)$$

Note that only for $m = 1$ does (13) become a *necessary* condition: If $\eta_2 > 2\eta_1$, then $f_1$ will always achieve better risk than an REC of $f_1$ and $f_2$. If a training method generates classifiers $f_1, ..., f_M$ with risks: $\eta_1 < 1$ and $\eta_i = 1 \; \forall i \in \{2, ..., M\}$, i.e., only the first classifier is somewhat robust and the remaining $M - 1$ classifiers are compromised (such as BAT), the lower bound in (11) reduces to:

$$\eta(\boldsymbol{\alpha}) \geq \min\left\{\eta_1, \frac{1}{M}\right\} \qquad (14)$$

implying the *necessary* condition $M \geq \lceil \eta_1^{-1} \rceil$ for RECs constructed from $\mathcal{F}$ to achieve better risk than $f_1$. Note: the fact that this condition is violated by (Pinot et al., 2020) hints to the existence of strong attacks that can break it (Zhang et al., 2022; Dbouk & Shanbhag, 2022).

## 4.4 Optimal Sampling

In this section, we leverage Proposition 1 to extend the results in Section 4.2 to provide a theoretically optimal and efficient solution for computing the optimal sampling probability (OSP) algorithm (Algorithm 1) for $M > 3$.

In practice, we do not know the true data distribution $\mathcal{D}$. Instead, we are provided a training set $z_1, ..., z_n$, assumed to be sampled i.i.d. from $\mathcal{D}$. Given the training set, and a fixed collection of classifiers $\mathcal{F}$, we wish to find the optimal sampling probability:

$$\boldsymbol{\alpha}^* = \underset{\boldsymbol{\alpha} \in \Delta_M}{\arg\min} \; \hat{\eta}(\boldsymbol{\alpha}) = \underset{\boldsymbol{\alpha} \in \Delta_M}{\arg\min} \; \frac{1}{n} \sum_{j=1}^n \left( \underset{\boldsymbol{\delta} \in \mathcal{S}}{\arg\max} \sum_{i=1}^M \alpha_i \mathbb{1}\left\{ f_i(\mathbf{x}_j + \boldsymbol{\delta}) \neq y_j \right\} \right) \qquad (15)$$

Note that the empirical adversarial risk $\hat{\eta}$ is also piece-wise linear and convex in $\boldsymbol{\alpha}$, and hence all our theoretical results apply naturally. In order to numerically solve (15), we first require access to an adversarial attack oracle (attack) for RECs that solves $\forall \mathcal{S}, \mathcal{F}, z$, and $\boldsymbol{\alpha}$:

$$\mathsf{attack}\,(\mathcal{F}, \mathcal{S}, \boldsymbol{\alpha}, z) = \underset{\boldsymbol{\delta} \in \mathcal{S}}{\arg\max} \sum_{i=1}^M \alpha_i \mathbb{1}\left\{ f_i(\mathbf{x} + \boldsymbol{\delta}) \neq y \right\} \qquad (16)$$

Using the oracle attack, Algorithm 1 updates its solution iteratively given the adversarial error-rate of each classifier over the training set. The projection operator $\Pi_{\Delta_M}$ in Line (15) of Algorithm 1 ensures that the solution is a valid p.m.f.. Wang & Carreira-Perpiñán (2013) provide a simple and exact method for computing $\Pi_{\Delta_M}$. Finally, we state the following result on the optimality of OSP:

**Theorem 3.** *The OSP algorithm output $\boldsymbol{\alpha}_T$ satisfies:*

$$0 \leq \hat{\eta}(\boldsymbol{\alpha}_T) - \hat{\eta}(\boldsymbol{\alpha}^*) \leq \frac{\|\boldsymbol{\alpha}^{(1)} - \boldsymbol{\alpha}^*\|_2^2 + Ma^2 \sum_{t=1}^T t^{-2}}{2a \sum_{t=1}^T t^{-1}} \xrightarrow[T \to \infty]{} 0 \qquad (17)$$

*for all initial conditions $\boldsymbol{\alpha}^{(1)} \in \Delta_M$, $a > 0$, where $\boldsymbol{\alpha}^*$ is a global minimum.*

Theorem 3 follows from a direct application of the classic convergence result of the projected subgradient method for constrained convex minimization (Shor (2012)). The optimality of OSP relies on the existence of an attack oracle for (16) which may not always exist. However, attack algorithms

such as ARC (Dbouk & Shanbhag (2022)) were found to yield good results in the common setting of differentiable classifiers and $\ell_p$-restricted adversaries.

---

**Algorithm 1** The Optimal Sampling Probability (OSP) Algorithm for Randomized Ensembles

1: **Input:** classifiers $\mathcal{F} = \{f_i\}_{i=1}^M$, perturbation set $\mathcal{S}$, attack algorithm attack, training set $\{z_j\}_{j=1}^n$, initial step-size $a > 0$, and number of iterations $T \geq 1$.
2: **Output:** optimal sampling probability $\boldsymbol{\alpha}^*$.
3: initialize $\boldsymbol{\alpha}^{(1)} \in \Delta_M, \eta_{\text{best}} \leftarrow 1$
4: /* we find that $\boldsymbol{\alpha}^{(1)} = \left[\frac{1}{M} \ldots \frac{1}{M}\right]^\top$ performs well
5: **for** $t \in \{1, ..., T\}$ **do**
6:    $\mathbf{g} \leftarrow \mathbf{0}, a_t \leftarrow \frac{a}{t}$
7:    **for** $j \in \{1, ..., n\}$ **do**
8:       $\boldsymbol{\delta}_j \leftarrow \text{attack}\left(\mathcal{F}, \mathcal{S}, \boldsymbol{\alpha}^{(t)}, z_j\right)$
9:       $\forall i \in [M]: g_i \leftarrow g_i + \mathbb{1}\{f_i(\mathbf{x}_j + \boldsymbol{\delta}_j) \neq y_j\}$
10:    **end for**
11:    $\mathbf{g} \leftarrow \frac{1}{n}\mathbf{g}$       ▷ sub-gradient of $\eta(\boldsymbol{\alpha}^{(t)})$
12:    $\eta^{(t)} \leftarrow \mathbf{g}^\top \boldsymbol{\alpha}^{(t)}$       ▷ $\eta(\boldsymbol{\alpha}^{(t)})$
13:    **if** $\eta^{(t)} \leq \eta_{\text{best}}$ **then** $t_{\text{best}} \leftarrow t, \eta_{\text{best}} \leftarrow \eta^{(t)}$
14:    /* projection-update step
15:    $\boldsymbol{\alpha}^{(t+1)} \leftarrow \Pi_{\Delta_M}\left(\boldsymbol{\alpha}^{(t)} - a_t\mathbf{g}\right)$
16: **end for**
17: **return** $\boldsymbol{\alpha}^{(t_{\text{best}})}$

---

**Algorithm 2** The Boosting Algorithm for Robust Randomized Ensembles (BARRE)

1: **Input:** Number of classifiers $M$, perturbation set $\mathcal{S}$, training set $\{z_j\}_{j=1}^n$, learning rate $\rho$, mini-batch size $B$, number of epochs $E$, OSP frequency $E_o$, OSP number of iterations $T_o$.
2: **Output:** Robust randomized ensemble classifier $(\mathcal{F}, \boldsymbol{\alpha})$
3: initialize $\boldsymbol{\theta}_0 \in \Theta, \mathcal{F} \leftarrow \varnothing$
4: **for** $m \in \{1, ..., M\}$ **do**
5:    $\boldsymbol{\theta}_m \leftarrow \boldsymbol{\theta}_{m-1}, \mathcal{F} \leftarrow \mathcal{F} \cup \{f(\cdot|\boldsymbol{\theta}_m)\}, \boldsymbol{\alpha} \leftarrow \left[\frac{1}{m} \ldots \frac{1}{m}\right]^\top$
6:    **for** $e \in \{1, ..., E\}$ **do**
7:       **for** mini-batch $\{z_b\}_b^B$ **do**
8:          compute $\forall b \in [B]: \boldsymbol{\delta}_b \leftarrow \text{attack}(\mathcal{F}, \mathcal{S}, \boldsymbol{\alpha}, z_b)$
9:          update $\boldsymbol{\theta}_m$ via SGD:

$$\boldsymbol{\theta}_m \leftarrow \boldsymbol{\theta}_m - \frac{\rho}{B}\sum_{b=1}^B \nabla_{\boldsymbol{\theta}_m} l\left(\tilde{f}(\mathbf{x}_b + \boldsymbol{\delta}_b | \boldsymbol{\theta}_m), y_b\right)$$

10:    **end for**
11:    /* update $\boldsymbol{\alpha}$ every $E_o$ epochs
12:    **if** $e \mod E_o = 0$ **then** $\boldsymbol{\alpha} \leftarrow \text{OSP}(\mathcal{F}, \mathcal{S}, \{z_j\}_{j=1}^n, T_o)$
13:    **end for**
14: **end for**
15: **return** $\mathcal{F}, \boldsymbol{\alpha}$

---

## 5   A ROBUST BOOSTING ALGORITHM FOR RANDOMIZED ENSEMBLES

Inspired by BAT (Pinot et al., 2020) and MRBoost (Zhang et al., 2022), we leverage our results in Section 4 and propose BARRE: a unified **B**oosting **A**lgorithm for **R**obust **R**andomized **E**nsembles described in Algorithm 2. Given a dataset $\{z_j\}_{j=1}^n$ and an REC attack algorithm attack, BARRE iteratively trains a set of parametric classifiers $f(\cdot|\boldsymbol{\theta}_1), ..., f(\cdot|\boldsymbol{\theta}_M)$ such that the adversarial risk of the corresponding REC is minimized. The first iteration of BARRE reduces to standard AT (Madry et al., 2018). Doing so typically guarantees that the first classifier achieves the lowest adversarial risk and $\eta(\boldsymbol{\alpha}^*) \leq \eta_1$, i.e., Theorem 3 ensures the REC is *no worse* than single model AT.

In each iteration $m \geq 2$, BARRE initializes the $m$-th classifier $f(\cdot|\boldsymbol{\theta}_m)$ with $\boldsymbol{\theta}_m = \boldsymbol{\theta}_{m-1}$. The training procedure alternates between updating the parameters $\boldsymbol{\theta}_m$ via SGD using adversarial samples of the current REC and solving for the optimal sampling probability $\boldsymbol{\alpha}^* \in \Delta_m$ via OSP. Including $f(\cdot|\boldsymbol{\theta}_m)$ in the attack (Line (8)) is crucial, as it ensures that the robustness of $f(\cdot|\boldsymbol{\theta}_m)$ is not completely compromised, thereby improving the bounds in Theorem 2. Note that for iterations $m \leq 3$, we replace the OSP procedure in Line (12) with a simplified search over a finite set of candidate solutions (see Section 4.2 and Appendix C.1).

### 5.1   EXPERIMENTAL RESULTS

In this section, we validate the effectiveness of BARRE in constructing robust RECs.

**Setup**. Per standard practice, we focus on defending against $\ell_\infty$ norm-bounded adversaries. We report results for three network architectures with different complexities: ResNet-20 (He et al., 2016), MobileNetV1 (Howard et al., 2017), and ResNet-18, across CIFAR-10 and CIFAR-100 datasets (Krizhevsky et al., 2009). Computational complexity is measured via the number of floating-point operations (FLOPs) required per inference. To ensure a fair comparison across different baselines, we use the same hyper-parameter settings detailed in Appendix D.1.

**Attack Algorithm**. For all our robust evaluations, we will adopt the state-of-the-art ARC algorithm (Dbouk & Shanbhag, 2022) which can be used for both RECs and single models. Specifically, we shall use a slightly modified version that achieves better results in the equiprobable sampling setting (see Appendix D.3). For training with BARRE, we adopt adaptive PGD (Zhang et al., 2022) for better generalization performance (see Appendix D.4).

**Benefit of Randomization**. Table 1 demonstrates that BARRE can successfully construct RECs that achieve competitive robustness (within $\sim 0.5\%$) compared to MRBoost-trained deterministic

Table 1: Comparison between BARRE and MRBoost across different network architectures and ensemble sizes on CIFAR-10. Robust accuracy is measured against an $\ell_\infty$ norm-bounded adversary using ARC with $\epsilon = 8/255$.

| Network | Method | $M = 1$ | | | $M = 2$ | | | $M = 3$ | | | $M = 4$ | | |
|---|---|---|---|---|---|---|---|---|---|---|---|---|---|
| | | $A_{\text{nat}}$ | $A_{\text{rob}}$ | FLOPs | $A_{\text{nat}}$ | $A_{\text{rob}}$ | FLOPs | $A_{\text{nat}}$ | $A_{\text{rob}}$ | FLOPs | $A_{\text{nat}}$ | $A_{\text{rob}}$ | FLOPs |
| ResNet-20 | MRBoost | 73.18 | 41.99 | 81 M | 75.22 | 44.68 | 162 M | 76.13 | 46.09 | 243 M | 76.96 | 46.34 | 324 M |
| | BARRE | | | | 74.63 | 44.38 | 81 M | 75.55 | 45.41 | 81 M | 75.95 | 46.44 | 81 M |
| MobileNetV1 | MRBoost | 79.01 | 46.22 | 312 M | 80.19 | 48.58 | 624 M | 79.79 | 49.39 | 936 M | 80.14 | 49.36 | 1.2 B |
| | BARRE | | | | 79.58 | 48.32 | 312 M | 79.53 | 48.75 | 312 M | 79.54 | 49.38 | 312 M |
| ResNet-18 | MRBoost | 80.96 | 48.7 | 1.1 B | 83.90 | 50.72 | 2.2 B | 85.07 | 52.15 | 3.3 B | 85.07 | 52.15 | 4.4 B |
| | BARRE | | | | 82.66 | 50.51 | 1.1 B | 83.40 | 51.57 | 1.1 B | 83.54 | 52.13 | 1.1 B |

ensembles, across three different network architectures on CIFAR-10. The benefit of randomization can be seen for $M \geq 2$, as we obtain *massive* $2 - 4\times$ savings in compute requirements. Note that both methods have the same[4] memory footprint. These observations are further corroborated by CIFAR-100 experiments in Appendix D.5.

Table 2: Comparison between BARRE and other methods at constructing robust randomized ensemble classifiers across various network architectures and datasets. Robust accuracy is measured against an $\ell_\infty$ norm-bounded adversary using ARC with $\epsilon = 8/255$.

| Network | FLOPs | Method | Size $M$ | CIFAR-10 | | CIFAR-100 | |
|---|---|---|---|---|---|---|---|
| | | | | $A_{\text{nat}}$ [%] | $A_{\text{rob}}$ [%] | $A_{\text{nat}}$ [%] | $A_{\text{rob}}$ [%] |
| ResNet-20 | 81 M | AT | $M = 1$ | 73.18 | 41.99 | 38.34 | 17.69 |
| | | IAT | $M = 5$ | 73.90 | 45.77 | 38.57 | 19.65 |
| | | MRBoost | $M = 5$ | 75.89 | 46.66 | 41.69 | 21.04 |
| | | BARRE | $M = 5$ | **76.28** | **47.35** | **41.86** | **21.11** |
| MobileNetV1 | 312 M | AT | $M = 1$ | 79.01 | 46.22 | 51.87 | 23.45 |
| | | IAT | $M = 5$ | 78.89 | 49.57 | 51.41 | 25.74 |
| | | MRBoost$^\dagger$ | $M = 5$ | 76.70 | 48.05 | 50.14 | 24.76 |
| | | MRBoost | $M = 5$ | 78.65 | 48.91 | **52.96** | 25.95 |
| | | BARRE | $M = 5$ | **79.55** | **49.91** | 52.95 | **27.53** |
| ResNet-18 | 1.1 B | AT | $M = 1$ | 80.96 | 48.72 | 53.85 | 24.15 |
| | | IAT | $M = 4$ | 80.99 | 51.43 | 54.30 | 26.73 |
| | | MRBoost$^\dagger$ | $M = 4$ | 83.13 | 51.82 | 51.06 | 24.04 |
| | | MRBoost | $M = 4$ | 83.13 | 51.82 | 52.04 | 25.65 |
| | | BARRE | $M = 4$ | **83.54** | **52.13** | **54.63** | **26.93** |

† result obtained assuming equiprobable sampling instead of using OSP

**BARRE vs. Other Methods**. Due to the lack of dedicated randomized ensemble defenses, we establish baselines by constructing RECs from both MRBoost and independently adversarially trained (IAT) models. We use OSP (Algorithm 1) to find the optimal sampling probability for each REC. All RECs share the same first classifier $f_1$, which is adversarially trained. Doing so ensures a fair comparison, and guarantees all the methods cannot be worse than AT. Table 2 provides strong evidence that BARRE outperforms both IAT and MRBoost for both CIFAR-10 and CIFAR-100 datasets. Interestingly, we find that MRBoost ensembles can be quite ill-suited for RECs. This can be seen for MobileNetV1, where the optimal sampling probability obtained was $\boldsymbol{\alpha}^* = [0.25 \ 0.25 \ 0.25 \ 0.25 \ 0]^\top$, i.e., the REC completely disregards the last classifier. In contrast, BARRE-trained RECs utilize all members of the ensemble.

## 6 CONCLUSION

We have demonstrated both theoretically and empirically that robust randomized ensemble classifiers (RECs) are realizable. Theoretically, we derive the robustness limits of RECs, necessary and sufficient conditions for them to be useful, and efficient methods for finding the optimal sampling probability. Empirically, we propose BARRE, a new boosting algorithm for constructing robust RECs and demonstrate its effectiveness at defending against strong $\ell_\infty$ norm-bounded adversaries.

---

[4]ignoring the negligible memory overhead of storing $\boldsymbol{\alpha}$

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

# A    THEORETICAL RATIONALE FOR BARRE

In this section, we expand on Section 5.1 and provide a more detailed rationale behind the steps in BARRE.

Given a dataset $\{z_j\}_{j=1}^n$ and an REC attack algorithm attack, BARRE iteratively trains a set of parametric classifiers $f(\cdot|\boldsymbol{\theta}_1)$, ..., $f(\cdot|\boldsymbol{\theta}_M)$ in a boosting fashion. Note that the optimality of OSP (Theorem 3) implies that a BARRE-trained REC is guaranteed to achieve better or the same performance than the BEST performing member of the ensemble. This explains the choice of boosting for BARRE, since the first iteration reduces to standard AT, the most effective method to date for generating robust classifiers. Thus, BARRE trained RECs are guaranteed to perform better than AT.

In each iteration $m \geq 2$, BARRE initializes the $m$-th classifier $f(\cdot|\boldsymbol{\theta}_m)$ with $\boldsymbol{\theta}_m = \boldsymbol{\theta}_{m-1}$. The training procedure alternates between updating the parameters $\boldsymbol{\theta}_m$ via SGD using adversarial samples of the current REC and solving for the optimal sampling probability $\boldsymbol{\alpha}^* \in \Delta_m$ via OSP. Including $f(\cdot|\boldsymbol{\theta}_m)$ in the attack (Line (8)) is crucial, as it ensures that the robustness of $f(\cdot|\boldsymbol{\theta}_m)$ is not completely compromised, thereby improving the bounds in Theorem 2.

Furthermore, the rationale behind the sequence of steps in BARRE can be better understood using Theorem 1 (for the case of $M = 2$). Theorem 1 states that the optimal REC adversarial risk would be $\eta(\boldsymbol{\alpha}^*) = \frac{1}{2}(\eta_1 + \eta_2 - \mathbb{P}\{z \in \mathcal{R}_1\})$ (assuming (8) is met), therefore it is equally important to minimize both $\eta$'s and maximize $\mathbb{P}\{z \in \mathcal{R}_1\}$. BARRE does so by initially adversarially training a robust classifier $f_1$ (minimizing $\eta_1$), then training $f_2$ (initialized from $f_1$ to minimizes $\eta_2$) on the adversarial examples of the REC of $f_1$ and $f_2$. Doing so increases $\mathbb{P}\{z \in \mathcal{R}_1\}$ while maintaining $\eta_2$ as low as possible.

## B  OMITTED PROOFS AND DERIVATIONS

### B.1  PROOF OF PROPOSITION 1

We provide the proof of Proposition 1 (restated below):

**Proposition** (Restated). *For any $\mathcal{F} = \{f_i\}_{i=1}^M$, perturbation set $\mathcal{S} \subset \mathbb{R}^d$, and data distribution $\mathcal{D}$, the adversarial risk $\eta$ is a piece-wise linear convex function $\forall \boldsymbol{\alpha} \in \Delta_M$. Specifically, $\exists K \in \mathbb{N}$ configurations $\mathcal{U}_k \subseteq \{0,1\}^M \; \forall k \in [K]$ and p.m.f. $\mathbf{p} \in \Delta_K$ such that:*

$$\eta(\boldsymbol{\alpha}) = \sum_{k=1}^K \left( p_k \cdot \max_{\mathbf{u} \in \mathcal{U}_k} \left\{ \mathbf{u}^\top \boldsymbol{\alpha} \right\} \right) \tag{18}$$

*Proof.* Consider having one data-point $\mathbf{z} \in \mathbb{R}^d \times [C]$, then for any $\boldsymbol{\delta} \in \mathcal{S}$ and $\boldsymbol{\alpha} \in \Delta_M$ we have:

$$r(\mathbf{z}, \boldsymbol{\delta}, \boldsymbol{\alpha}) = \sum_{i=1}^M \alpha_i \mathbb{1}\left\{ f_i(\mathbf{x} + \boldsymbol{\delta}) \neq y \right\} = \mathbf{u}^\top \boldsymbol{\alpha} \tag{19}$$

where $\mathbf{u} \in \{0,1\}^M$ such that $u_i = 1$ if and only if $\boldsymbol{\delta}$ is adversarial to $f_i$ at $\mathbf{z}$. Since $\mathbf{u}$ is independent of $\boldsymbol{\alpha}$, we thus obtain a many-to-one mapping from $\boldsymbol{\delta} \in \mathcal{S}$ to $\mathbf{u} \in \{0,1\}^M$. Therefore, for any $\boldsymbol{\alpha}$ and $\mathbf{z}$, we can always decompose the perturbation set $\mathcal{S}$, i.e., $\mathcal{S} = \mathcal{G}_1 \cup ... \cup \mathcal{G}_n$, into $n \leq 2^M$ subsets, such that: $\forall \boldsymbol{\delta} \in \mathcal{G}_j : r(\mathbf{z}, \boldsymbol{\delta}, \boldsymbol{\alpha}) = \boldsymbol{\alpha}^\top \mathbf{u}_j$ for some binary vector $\mathbf{u}_j$ independent of $\boldsymbol{\alpha}$. Let $\mathcal{U} = \{\mathbf{u}_j\}_{j=1}^n$ be the collection of these vectors, then we can write:

$$\max_{\boldsymbol{\delta} \in \mathcal{S}} r(\mathbf{z}, \boldsymbol{\delta}, \boldsymbol{\alpha}) = \max_{\boldsymbol{\delta} \in \mathcal{G}_1 \cup ... \cup \mathcal{G}_n} r(\mathbf{z}, \boldsymbol{\delta}, \boldsymbol{\alpha}) = \max_{j \in [n]} \left\{ \max_{\boldsymbol{\delta} \in \mathcal{G}_j} r(\mathbf{z}, \boldsymbol{\delta}, \boldsymbol{\alpha}) \right\} = \max_{\mathbf{u} \in \mathcal{U}} \left\{ \mathbf{u}^\top \boldsymbol{\alpha} \right\} \tag{20}$$

The vectors $\{\mathbf{u}_j\}_{j=1}^n$ define a unique classifier and data-point configuration that is independent of the sampling probability. The function $\max_{\boldsymbol{\delta}} r$ is thus convex and piece-wise linear in $\boldsymbol{\alpha}$.

Partitioning the data-point space $\mathcal{R} \subseteq \mathbb{R}^d \times [C]$ into $K$ subsets $\mathcal{R} = \mathcal{R}_1 \cup ... \cup \mathcal{R}_K$ such that all the data-points $\mathbf{z} \in \mathcal{R}_k$ share the same set "configuration" $\mathcal{U}_k$, we obtain:

$$\begin{aligned}
\eta(\boldsymbol{\alpha}) &= \mathbb{E}_{\mathbf{z} \sim \mathcal{D}} \left[ \max_{\boldsymbol{\delta} \in \mathcal{S}} \sum_{i=1}^M \alpha_i \mathbb{1}\left\{ f_i(\mathbf{x} + \boldsymbol{\delta}) \neq y \right\} \right] \\
&= \int_{\mathbf{z} \in \mathcal{R}} p_z(\mathbf{z}) \cdot \max_{\boldsymbol{\delta} \in \mathcal{S}} r(\mathbf{z}, \boldsymbol{\delta}, \boldsymbol{\alpha}) \, d\mathbf{z} \\
&= \sum_{k=1}^K \int_{\mathbf{z} \in \mathcal{R}_k} p_z(\mathbf{z}) \cdot \max_{\boldsymbol{\delta} \in \mathcal{S}} r(\mathbf{z}, \boldsymbol{\delta}, \boldsymbol{\alpha}) \, d\mathbf{z} \\
&= \sum_{k=1}^K \int_{\mathbf{z} \in \mathcal{R}_k} p_z(\mathbf{z}) \cdot \left( \max_{\mathbf{u} \in \mathcal{U}_k} \left\{ \mathbf{u}^\top \boldsymbol{\alpha} \right\} \right) d\mathbf{z} \\
&= \sum_{k=1}^K \left( \max_{\mathbf{u} \in \mathcal{U}_k} \left\{ \mathbf{u}^\top \boldsymbol{\alpha} \right\} \cdot \int_{\mathbf{z} \in \mathcal{R}_k} p_z(\mathbf{z}) \, d\mathbf{z} \right) \\
&= \sum_{k=1}^K \left( p_k \cdot \max_{\mathbf{u} \in \mathcal{U}_k} \left\{ \mathbf{u}^\top \boldsymbol{\alpha} \right\} \right)
\end{aligned} \tag{21}$$

where the total size of the partition $K$ is finite (exponential in the size $M$) and $\mathbf{p} \in \Delta_K$ such that $p_k = \mathbb{P}\{\mathbf{z} \in \mathcal{R}_k\}$. Finally, $\eta$ is convex and piece-wise linear in $\boldsymbol{\alpha}$ since the summation of convex and piece-wise linear functions is also convex and piece-wise linear. $\square$

### B.2  PROOF OF THEOREM 1

First, we state and prove this useful lemma:

**Lemma 1.** *Let $h : \mathbb{R} \to \mathbb{R}$ be a convex piece-wise linear, hence sub-differentiable, function of the form:*

$$h(x) = \max\{a_1 x + b_1, a_2 x + b_2\} + a_3 x + b_3 \tag{22}$$

*such that $a_1 < a_2$. We wish to minimize $h$ over $x \in [c, d]$ where $c \leq y \leq d$, and $y$ is the intersection point $\frac{b_2 - b_1}{a_1 - a_2}$.*

*Then, the optimal value $x^*$ that minimizes $h(x)$ in (22), is given by*

$$x^* = \begin{cases} y, & \text{if } a_3 \in (-a_2, -a_1) \\ c, & \text{if } a_3 \geq -a_1 \\ d, & \text{if } a_3 \leq -a_2 \end{cases}$$

*Note: only in the first case is the solution unique.*

*Proof.* From constrained convex optimization (Boyd et al. (2004); Shor (2012)), we know that $x^*$ is the minimizer of $h$ over $[c, d]$ if there exists a sub-gradient $g \in \partial h(x^*)$ such that:

$$g \cdot (x - x^*) \geq 0 \quad \forall x \in [c, d] \tag{23}$$

For $x \neq y$, $h$ is differentiable with $\nabla h = a_3 + a_1$ (if $x < y$) or $\nabla h = a_3 + a_2$ (if $x > y$), and for $x = y$ the sub-differential is given by $\partial h(y) = \{a_3 + \beta a_1 + (1 - \beta) a_2 : \beta \in [0, 1]\}$.

If $a_3 \in (-a_2, -a_1)$, then $\exists \beta \in [0, 1]$ such that $a_3 + \beta a_1 + (1 - \beta) a_2 = 0$, and thus $0 \in \partial h(y)$, which is a sufficient condition for global minimization, thus $x^* = y$. Furthermore, $x^* = y$ is unique, since $\forall x \neq y$, we will have $\nabla h = a_1 + a_3 < 0$ (if $x < y$) or $\nabla h = a_2 + a_3 > 0$ (if $x > y$) which in both cases implies $\forall z \neq y \ \exists x \in [c, d]$ such that $\nabla h(z)(x - z) < 0$.

If $a_3 \notin (-a_2, -a_1)$, then either $a_3 \geq -a_1$ or $a_3 \leq -a_2$. If $a_3 \geq -a_1$, then $a_1 + a_3 = \nabla h(c) \geq 0$, which implies that: $(a_1 + a_3)(x - c) \geq 0 \ \forall x \in [c, d]$, hence $x^* = c$. Otherwise if $a_3 \leq -a_2$, then $a_2 + a_3 = \nabla h(d) \leq 0$, which implies that: $(a_2 + a_3)(x - d) \geq 0 \ \forall x \in [c, d]$, hence $x^* = d$. $\qquad \square$

We now provide the proof of Theorem 1 (restated below):

**Theorem** (Restated). *For any two classifiers $f_1$ and $f_2$ with individual adversarial risks $\eta_1$ and $\eta_2$, respectively, subject to a perturbation set $\mathcal{S} \subset \mathbb{R}^d$ and data distribution $\mathcal{D}$, if:*

$$\mathbb{P}\{\mathbf{z} \in \mathcal{R}_1\} > |\eta_1 - \eta_2| \tag{24}$$

*where:*

$$\mathcal{R}_1 = \{\mathbf{z} \in \mathbb{R}^d \times [C] : \mathcal{S}_1(\mathbf{z}) \neq \varnothing, \mathcal{S}_2(\mathbf{z}) \neq \varnothing, \mathcal{S}_1(\mathbf{z}) \cap \mathcal{S}_2(\mathbf{z}) = \varnothing\} \tag{25}$$

*then the optimum sampling probability $\boldsymbol{\alpha}^* = (1/2, 1/2)^\top$ uniquely minimizes $\eta(\boldsymbol{\alpha})$ resulting in $\eta(\boldsymbol{\alpha}^*) = \frac{1}{2}(\eta_1 + \eta_2 - \mathbb{P}\{\mathbf{z} \in \mathcal{R}_1\})$. Otherwise, $\boldsymbol{\alpha}^* \in \{\mathbf{e}_1, \mathbf{e}_2\}$ minimizes $\eta(\boldsymbol{\alpha})$, where $\mathbf{e}_i$s are the standard basis vectors of $\mathbb{R}^2$.*

*Proof.* We know that, for $M = 2$, the adversarial risk $\eta$ can be re-written $\forall \boldsymbol{\alpha} \in \Delta_2$:

$$\eta(\boldsymbol{\alpha}) = p_1 \cdot \max\{\alpha_1, \alpha_2\} + p_2 \cdot 1 + p_3 \cdot \alpha_1 + p_4 \cdot \alpha_2 + p_5 \cdot 0 \tag{26}$$

where $p_k = \mathbb{P}\{\mathbf{z} \in \mathcal{R}_k\}$, and the regions $\{\mathcal{R}_k\}_{k=1}^K$ partition the input space $\mathbb{R}^d \times [C]$ as follows:

$$\mathcal{R}_1 = \{\mathbf{z} \in \mathbb{R}^d \times [C] : \mathcal{S}_1(\mathbf{z}) \neq \varnothing, \mathcal{S}_2(\mathbf{z}) \neq \varnothing, \mathcal{S}_1(\mathbf{z}) \cap \mathcal{S}_2(\mathbf{z}) = \varnothing\}$$
$$\mathcal{R}_2 = \{\mathbf{z} \in \mathbb{R}^d \times [C] : \mathcal{S}_1(\mathbf{z}) \cap \mathcal{S}_2(\mathbf{z}) \neq \varnothing\}$$
$$\mathcal{R}_3 = \{\mathbf{z} \in \mathbb{R}^d \times [C] : \mathcal{S}_1(\mathbf{z}) \neq \varnothing, \mathcal{S}_2(\mathbf{z}) = \varnothing\} \tag{27}$$
$$\mathcal{R}_4 = \{\mathbf{z} \in \mathbb{R}^d \times [C] : \mathcal{S}_1(\mathbf{z}) = \varnothing, \mathcal{S}_2(\mathbf{z}) \neq \varnothing\}$$
$$\mathcal{R}_5 = \{\mathbf{z} \in \mathbb{R}^d \times [C] : \mathcal{S}_1(\mathbf{z}) = \varnothing, \mathcal{S}_2(\mathbf{z}) = \varnothing\}$$

Using $\alpha_1 = 1 - \alpha_2 = \alpha$, we have $\forall \alpha \in [0, 1]$:

$$\eta\left((\alpha, 1 - \alpha)^\top\right) = h(\alpha) = p_1 \cdot \max\{\alpha, 1 - \alpha\} + (p_3 - p_4) \cdot \alpha + p_2 + p_4 \tag{28}$$

where we wish to find $\alpha^* \in [0, 1]$ that minimizes $h(\alpha)$. Applying Lemma 1 with:

$$a_1 = -p_1, \ b_1 = p_1, \ a_2 = p_1, \ b_2 = 0, \ a_3 = p_3 - p_4, \ b_3 = p_2 + p_4 \tag{29}$$

and utilizing $\eta_1 = \eta(\mathbf{e}_1) = p_1 + p_2 + p_3$ and $\eta_2 = \eta(\mathbf{e}_2) = p_1 + p_2 + p_4$, yields the main result. $\qquad \square$

### B.3 PROOF OF COROLLARY 1

We provide the proof of Corollary 1 (restated below):

**Corollary.** *For any two classifiers $f_1$ and $f_2$ with individual adversarial risks $\eta_1$ and $\eta_2$, respectively, perturbation set $\mathcal{S}$, and data distribution $\mathcal{D}$:*

$$\min_{\boldsymbol{\alpha} \in \Delta_2} \eta(\boldsymbol{\alpha}) = \eta(\boldsymbol{\alpha}^*) \geq \min\left\{\frac{1}{2}\max\{\eta_1, \eta_2\}, \min\{\eta_1, \eta_2\}\right\}. \tag{30}$$

*Proof.* From Theorem 1, we have that:

$$\eta(\boldsymbol{\alpha}^*) = \min\left\{\frac{1}{2}\left(\eta_1 + \eta_2 - \mathbb{P}\{\mathbf{z} \in \mathcal{R}_1\}\right), \min\{\eta_1, \eta_2\}\right\} \tag{31}$$

Using the tight upper bound on $\mathbb{P}\{\mathbf{z} \in \mathcal{R}_1\} \leq \min\{\eta_1, \eta_2\}$, we obtain the main result. $\qquad\square$

### B.4 PROOF OF THEOREM 2

#### B.4.1 USEFUL LEMMAS

We first state and prove a few useful lemmas that are vital for proving Theorem 2. While some lemmas are trivial and have been proven elsewhere, we nonetheless state their proofs for completeness.

**Lemma 2.** *Let $h : \mathbb{R}^n \to \mathbb{R}$ be a convex function, and $\mathcal{H}(\mathcal{X}) \subset \mathbb{R}^n$ be the convex hull of $\mathcal{X} = \{\mathbf{x}_1, ..., \mathbf{x}_d\}$ where $\{\mathbf{x}_i\}_{i=1}^d \in \mathbb{R}^n$, then there exists $\mathbf{x}_m \in \mathcal{X}$ such that:*

$$\max_{\mathbf{u} \in \mathcal{H}(\mathcal{X})} h(\mathbf{u}) = h(\mathbf{x}_m) \tag{32}$$

*Proof.* Let $\mathbf{u}$ be any arbitrary vector in $\mathcal{H}(\mathcal{X})$, that is $\exists \boldsymbol{\alpha} \in \Delta_d$:

$$\mathbf{u} = \sum_{i=1}^d \alpha_i \mathbf{x}_i \tag{33}$$

Let $m \in [d]$ such that $h(\mathbf{x}_m) \geq h(\mathbf{x}_i) \ \forall i \in [d]$. From the convexity of $h$, we upper bound $h(\mathbf{u})$ as follows:

$$h(\mathbf{u}) = h\left(\sum_{i=1}^d \alpha_i \mathbf{x}_i\right) \leq \sum_{i=1}^d \alpha_i h(\mathbf{x}_i) \leq \sum_{i=1}^d \alpha_i h(\mathbf{x}_m) = h(\mathbf{x}_m) \sum_{i=1}^d \alpha_i = h(\mathbf{x}_m) \tag{34}$$

Thus, (32) holds for any $\mathbf{u} \in \mathcal{H}(\mathcal{X})$. $\qquad\square$

**Lemma 3** (Redistribution Lemma). *$\forall p, q$ such that $0 \leq p \leq q \leq 1$, $\forall \boldsymbol{\alpha} \in \Delta_M$, and $\forall \mathcal{I}, \mathcal{J} \subseteq [M]$ such that $\mathcal{I} \notin \mathcal{J} \notin \mathcal{I}$ we have:*

$$p \cdot \max_{i \in \mathcal{I}}\{\alpha_i\} + q \cdot \max_{j \in \mathcal{J}}\{\alpha_j\} \geq p \cdot \max_{i \in \mathcal{I} \cup \mathcal{J}}\{\alpha_i\} + (q - p) \cdot \max_{j \in \mathcal{J}}\{\alpha_j\} + p \cdot \max_{k \in \mathcal{I} \cap \mathcal{J}}\{\alpha_k\} \tag{35}$$

*Proof.*

$$\begin{aligned}
p \cdot \max_{i \in \mathcal{I}}\{\alpha_i\} + q \cdot \max_{j \in \mathcal{J}}\{\alpha_j\} &= p \cdot \alpha_{i^*} + q \cdot \alpha_{j^*} \\
&= p \cdot (\alpha_{i^*} + \alpha_{j^*}) + (q - p) \cdot \alpha_{j^*} \\
&\overset{(a)}{=} p \cdot \left(\max_{i \in \mathcal{I} \cup \mathcal{J}}\{\alpha_i\} + \min\{\alpha_{i^*}, \alpha_{j^*}\}\right) + (q - p) \cdot \alpha_{j^*} \\
&\overset{(b)}{\geq} p \cdot \max_{i \in \mathcal{I} \cup \mathcal{J}}\{\alpha_i\} + (q - p) \cdot \alpha_{j^*} + p \cdot \max_{k \in \mathcal{I} \cap \mathcal{J}}\{\alpha_k\} \\
&= p \cdot \max_{i \in \mathcal{I} \cup \mathcal{J}}\{\alpha_i\} + (q - p) \cdot \max_{j \in \mathcal{J}}\{\alpha_j\} + p \cdot \max_{k \in \mathcal{I} \cap \mathcal{J}}\{\alpha_k\}
\end{aligned} \tag{36}$$

where (a) holds because the maximum over $\mathcal{I} \cup \mathcal{J}$ is either $\alpha_{i^*}$ or $\alpha_{j^*}$, and (b) holds since the smallest of the two maximizers cannot be smaller than the maximizer of the smaller set $\mathcal{I} \cap \mathcal{J}$. $\qquad\square$

**Lemma 4.** *Let $\{f_i\}_{i=1}^M$ be an arbitrary collection of $C$-ary classifiers with individual adversarial risks $\eta_i$ such that $0 < \eta_1 \leq ... \leq \eta_M \leq 1$. For any data distribution $\mathcal{D}$ and perturbation set $\mathcal{S}$ we have $\forall \boldsymbol{\alpha} \in \Delta_M$:*

$$\eta(\boldsymbol{\alpha}) \geq \sum_{i=1}^M \left( (\eta_i - \eta_{i-1}) \cdot \max_{j \in \{i,...,M\}} \{\alpha_j\} \right) \tag{37}$$

*where $\eta_0 \doteq 0$.*

*Proof.* From Proposition 1 we know that $\exists K \in \mathbb{N}$, $\mathbf{p} \in \Delta_K$, and $\mathcal{U}_k \subseteq \{0,1\}^M$ $\forall k \in [K]$ such that:

$$\eta(\boldsymbol{\alpha}) = \sum_{k=1}^K \left( p_k \cdot \max_{\mathbf{u} \in \mathcal{U}_k} \{\mathbf{u}^\top \boldsymbol{\alpha}\} \right) \tag{38}$$

Let $\mathcal{L}_k \subseteq [M]$ represent the set of classifier indices $i_1, ..., i_n$ that are active in the configuration $\mathcal{U}_k$, that is:

$$m \in \mathcal{L}_k \iff \exists \mathbf{v} \in \mathcal{U}_k \text{ such that } v_m = 1 \tag{39}$$

We then lower bound $\eta$ as follows:

$$\eta(\boldsymbol{\alpha}) = \sum_{k=1}^K \left( p_k \cdot \max_{\mathbf{u} \in \mathcal{U}_k} \{\mathbf{u}^\top \boldsymbol{\alpha}\} \right) \geq \sum_{k=1}^K \left( p_k \cdot \max_{i \in \mathcal{L}_k} \{\alpha_i\} \right) = \eta'(\boldsymbol{\alpha}) \tag{40}$$

The bound trivially holds, since the sum of positive numbers is always larger than any summand. It is noteworthy to point out that the RHS quantity $\eta'(\boldsymbol{\alpha})$ can be interpreted as the adversarial risk of an *auxiliary* set of classifiers $\mathcal{F}'$ with same individual risks $\{\eta_i\}$ such that for any $\mathbf{z} \in \mathbb{R}^d \times [C]$, the classifiers have no common adversarial perturbations, i.e.:

$$\bigcap_{i=1}^M \mathcal{S}'_i(\mathbf{z}) = \varnothing \tag{41}$$

and:

$$\eta'_i = \eta'(\mathbf{e}_i) = \sum_{k:i \in \mathcal{L}_k} p_k = \eta(\mathbf{e}_i) = \eta_i \tag{42}$$

Assume that the conditions of Lemma 3 are met by two terms in $\eta'$, i.e., $\exists k_1, k_2 \in [K]$ such that $\mathcal{L}_{k_1} \notin \mathcal{L}_{k_2} \notin \mathcal{L}_{k_1}$ and $p_{k_1} \leq p_{k_2}$, then we can apply the bound in Lemma 3 and obtain:

$$
\begin{aligned}
\eta'(\boldsymbol{\alpha}) - \sum_{k \in [K] \backslash \{k_1, k_2\}} \left( p_k \cdot \max_{i \in \mathcal{L}_k} \{\alpha_i\} \right) &= p_{k_1} \cdot \max_{i \in \mathcal{L}_{k_1}} \{\alpha_i\} + p_{k_2} \cdot \max_{i \in \mathcal{L}_{k_2}} \{\alpha_i\} \\
&\geq p_{k_1} \cdot \max_{i \in \mathcal{L}_{k_1} \cup \mathcal{L}_{k_2}} \{\alpha_i\} + (p_{k_2} - p_{k_1}) \cdot \max_{j \in \mathcal{L}_{k_2}} \{\alpha_j\} + p_{k_1} \cdot \max_{k \in \mathcal{L}_{k_1} \cap \mathcal{L}_{k_2}} \{\alpha_k\} \\
&= \eta''(\boldsymbol{\alpha}) - \sum_{k \in [K] \backslash \{k_1, k_2\}} \left( p_k \cdot \max_{i \in \mathcal{L}_k} \{\alpha_i\} \right)
\end{aligned}
\tag{43}
$$

where $\eta''(\boldsymbol{\alpha})$ is the modified ensemble adversarial risk. The application of Lemma 3 can be understood as a way to "re-distribute" the classifiers' adversarial vulnerabilities while preserving the adversarial risk identities $\forall i \in [M]$:

$$\eta_i = \eta'(\mathbf{e}_i) = \sum_{k:i \in \mathcal{L}_k} p_k = \eta''(\mathbf{e}_i) \tag{44}$$

The main idea of this proof is to keep applying Lemma 3 to the modified ensemble adversarial risks (if possible) to obtain a better lower bound. The process stops when the conditions are no longer met, and we obtain an adversarial risk $h(\boldsymbol{\alpha})$:

$$\eta'(\boldsymbol{\alpha}) \geq \eta''(\boldsymbol{\alpha}) \geq .. \geq h(\boldsymbol{\alpha}) = \sum_{k=1}^L \left( q_k \cdot \max_{j \in \mathcal{J}_k} \{\alpha_j\} \right) \tag{45}$$

Without loss of generality, we will assume that $\{\mathcal{J}_k\}$ are distinct and $q_k \neq 0$. Furthermore, since the conditions of Lemma 3 cannot be met by any two sets in $\{\mathcal{J}_k\}$, we must have (up to a re-ordering of the indices):

$$\mathcal{J}_L \subset \mathcal{J}_{L-1} \subset ... \subset \mathcal{J}_1 \subseteq [M] \tag{46}$$

We now make the following observations:

1. Due to (46), we have that $L \leq M$ and for all $i \in [M]$, $\exists m_i \in [L]$ such that:

$$\eta_i = \sum_{k:i\in\mathcal{J}_k} q_k = \sum_{k=1}^{m_i} q_k \tag{47}$$

2. Since $\{\eta_i\}$ are sorted, we get that $m_{i+1} = m_i + 1$ if $\eta_i < \eta_{i+1}$ or $m_{i+1} = m_i$ otherwise

3. $\mathcal{J}_1 = [M]$ since $\eta_1 \neq 0$

4. For any two consecutive sets $\mathcal{J}_k$ and $\mathcal{J}_{k+1}$, we can always find $n \geq 1$ indices from $[M]$ such that $\mathcal{J}_k = \mathcal{J}_{k+1} \cup \{i_1, ..., i_n\}$. The indices $i_1, ..., i_n$ are consecutive, share the same $m_i$ (i.e., $\eta_{i_l}$ is the same for all $l \in [n]$), and also satisfy:

$$\min_{l\in[n]}\{i_l\} = \max_{j\in\mathcal{J}_{k+1}}\{j\} + 1 \tag{48}$$

We first prove the lemma for the special case of distinct risks, i.e. $\eta_i < \eta_{i+1}$ $\forall i$.

**Special Case**. The risks are distinct, then we must have $L = M$, with every two consecutive sets $\mathcal{J}_k$ and $\mathcal{J}_{k+1}$ differing by one index. Therefore we have $\mathcal{J}_k = \mathcal{J}_{k+1} \cup \{k\}$ and $\mathcal{J}_{M+1} = \varnothing$. Furthermore, we will get $\eta_i - \eta_{i-1} = q_i$ $\forall i \in [M]$ with $\eta_0 = 0$. Thus we can write:

$$h(\boldsymbol{\alpha}) = \sum_{k=1}^{M}\left(q_k \cdot \max_{j\in\mathcal{J}_k}\{\alpha_j\}\right) = \sum_{i=1}^{M}\left((\eta_i - \eta_{i-1}) \cdot \max_{j\in\{i,...,M\}}\{\alpha_j\}\right) \tag{49}$$

**General Case**. For the general case we will have $L \leq M$ distinct risks $\eta_{i_1} < ... < \eta_{i_L}$ and $M - L$ repeated risks, where $i_1 = 1$. Thus we have $q_k = \eta_{i_k} - \eta_{i_{k-1}}$ $\forall k \in [L]$, and $\eta_{i_0} = \eta_0 = 0$ by definition. Using observations 3 and 4, we have that $\mathcal{J}_k = \{u_k, ..., M\}$ for some index $u_k \in [M]$, with $u_1 = 1$. Thus we have $u_{k+1} - u_k - 1 \geq 0$ to be the number of of consecutive repeated risks equal to $\eta_{i_k}$. Let $\{\mathcal{J}'_k\}$ be the $M - L$ index sets missing from $\{i \in [M] : \{i, ..., M\}\}$, then we have:

$$\begin{aligned}
h(\boldsymbol{\alpha}) &= \sum_{k=1}^{L}\left(q_k \cdot \max_{j\in\mathcal{J}_k}\{\alpha_j\}\right) \\
&= \sum_{k=1}^{L}\left((\eta_{i_{k+1}} - \eta_{i_k}) \cdot \max_{j\in\{u_k,...,M\}}\{\alpha_j\}\right) \\
&= \sum_{k=1}^{L}\left((\eta_{i_{k+1}} - \eta_{i_k}) \cdot \max_{j\in\{u_k,...,M\}}\{\alpha_j\}\right) + \sum_{k=1}^{M-L}\left(0 \cdot \max_{j\in\mathcal{J}'_k}\{\alpha_j\}\right) \\
&\overset{(a)}{=} \sum_{i=1}^{M}\left((\eta_i - \eta_{i-1}) \cdot \max_{j\in\{i,...,M\}}\{\alpha_j\}\right)
\end{aligned} \tag{50}$$

where (a) holds due to the fact $\eta_i - \eta_{i-1} = 0$ for all the merged $M - L$ terms. $\square$

**Lemma 5.** *Given a sequence $\{\gamma_i\}_{i=0}^{M}$ such that $0 = \gamma_0 < \gamma_1 \leq ... \leq \gamma_M \leq 1$, the vector $\boldsymbol{\alpha}^* = \left[\frac{1}{m} ... \frac{1}{m} 0 ... 0\right]^\top \in \Delta_M$ is a solution to the following minimization problem:*

$$\min_{\boldsymbol{\alpha}\in\Delta_M} h(\boldsymbol{\alpha}) = \min_{\boldsymbol{\alpha}\in\Delta_M} \sum_{i=1}^{M}\left((\gamma_i - \gamma_{i-1}) \cdot \max_{j\in\{i,...,M\}}\{\alpha_j\}\right) = \frac{\gamma_m}{m} \tag{51}$$

*where $\frac{\gamma_m}{m} \leq \frac{\gamma_i}{i}$, $\forall i \in [M]$.*

*Proof.* We know that $h$ is a piece-wise linear convex function over a closed and convex set, which implies the existence of a global minimizer.

Define the mapping $g : \Delta_M \to [0, 1]^M$ such that $\forall i \in [M]$:

$$g_i(\boldsymbol{\alpha}) = \max_{j \in \{i,...,M\}} \{\alpha_j\} - \max_{j \in \{i+1,...,M\}} \{\alpha_j\} \tag{52}$$

We can re-write the function $h$ via a simple re-arrangement to obtain:

$$h(\boldsymbol{\alpha}) = \sum_{i=1}^{M} \gamma_i \cdot \left( \max_{j \in \{i,...,M\}} \{\alpha_j\} - \max_{j \in \{i+1,...,M\}} \{\alpha_j\} \right) = \sum_{i=1}^{M} \gamma_i \cdot g_i(\boldsymbol{\alpha}) = \boldsymbol{\gamma}^\top g(\boldsymbol{\alpha}) \tag{53}$$

Define the decomposition over the probability simplex: $\Delta_M = \Delta_M^1 \cup \Delta_M^2 \cup ... \cup \Delta_M^{M!}$, where $\forall n \in [M!], \exists i_1, i_2, ..., i_M$ such that $\forall \boldsymbol{\alpha} \in \Delta_M^n$ we have:

$$\alpha_{i_1} \geq \alpha_{i_2} \geq \alpha_{i_3} \geq ... \geq \alpha_{i_M} \tag{54}$$

In other words, $\Delta_M^n$ is the set of all probability vectors that share the same sorting indices. Since we have $M!$ ways to arrange $M$ numbers, the size of the decomposition will be $M!$. We now make the following observations:

**1**. $\forall n$, $\Delta_M^n$ is a convex set. *quick proof*: Let $\boldsymbol{\alpha}, \boldsymbol{\beta} \in \Delta_M^n$, then $\exists i_1, i_2, ..., i_M$ such that $\alpha_{i_1} \geq ... \geq \alpha_{i_M}$ and $\beta_{i_1} \geq ... \geq \beta_{i_M}$. $\forall \lambda \in [0, 1]$ we have $\mathbf{q} = \lambda \boldsymbol{\alpha} + (1 - \lambda) \boldsymbol{\beta} \in \Delta_M$, since $\sum q_i = \sum \lambda \alpha_i + (1 - \lambda) \beta_i = 1$ and $q_i \geq 0$. We also have $\forall l \in [M - 1]$:

$$q_{i_l} = \lambda \alpha_{i_l} + (1 - \lambda) \beta_{i_l} \geq \lambda \alpha_{i_{l+1}} + (1 - \lambda) \beta_{i_{l+1}} = q_{i_{l+1}} \tag{55}$$

**2**. $\forall n$, $\exists \mathcal{P}^n = \{\mathbf{p}_1^n, ..., \mathbf{p}_M^n\} \subset \Delta_M^n$ such that $\Delta_M^n = \mathcal{H}(\mathcal{P}^n)$, where $\mathcal{H}(\mathcal{X})$ is the convex hull of the set of points $\mathcal{X}$. *quick proof*: Let $i_1, ..., i_M$ be the sorted indices associated with an arbitrary subset $\Delta_M^n$. Construct the $M$ probability vectors as follows: $\forall k \in [M]$ $p_{k,j}^n = \frac{1}{k}$ if $j \in \{i_1, ..., i_k\}$ else $p_{k,j}^n = 0$. It is easy to verify that $\mathbf{p}_k^n \in \Delta_M^n$, since $\sum_j p_{k,j}^n = k/k = 1$, and $p_{k,i_1}^n \geq ... \geq p_{k,i_M}^n$. Since $\Delta_M^n$ is convex (Claim 1), we thus have that $\mathcal{H}(\mathcal{P}^n) \subseteq \Delta_M^n$. What is left is to show that $\Delta_M^n \subseteq \mathcal{H}(\mathcal{P}^n)$, which can be established if we show that $\forall \boldsymbol{\alpha} \in \Delta_M^n, \exists \boldsymbol{\lambda} \in \Delta_M$ such that $\boldsymbol{\alpha} = \sum_k \lambda_k \mathbf{p}_k^n$. We shall prove it by construction, specifically define:

$$\lambda_k = k \cdot (\alpha_{i_k} - \alpha_{i_{k+1}}) \geq 0 \tag{56}$$

This induces a valid convex coefficient vector $\boldsymbol{\lambda}$, since $\sum_k \lambda_k = \sum_k (\alpha_{i_k} - \alpha_{i_{k+1}}) \cdot k = \sum_k \alpha_{i_k} = 1$. It is also easy to verify that $\alpha_{i_l} = \sum_k \lambda_k p_{k,i_l}^n$ for all indices $i_l \in [M]$, since:

$$\sum_{k=1}^{M} \lambda_k p_{k,i_l}^n = \frac{M}{M} \cdot (\alpha_{i_M} - 0) + \frac{M-1}{M-1} \cdot (\alpha_{i_{M-1}} - \alpha_{i_M}) + ... + \frac{l}{l} \cdot (\alpha_{i_l} - \alpha_{i_{l+1}}) = \alpha_{i_l} \tag{57}$$

by construction of $\boldsymbol{\lambda}$ and $\mathcal{P}^n$.

**3**. $\forall n$, the function $g$ is linear over $\boldsymbol{\alpha} \in \Delta_M^n$. *quick proof*: Define the maximum index $s(\boldsymbol{\alpha}, i) = \arg\max_{j \in \{i,...,M\}} \{\alpha_j\}$. By definition, $\boldsymbol{\alpha} \in \Delta_M^n$ implies that $s(i) = s(\boldsymbol{\alpha}, i)$ is independent of $\boldsymbol{\alpha}$. Therefore $\forall i \in [M]$ we have $g_i(\boldsymbol{\alpha}) = \alpha_{s(i)} - \alpha_{s(i+1)}$ with the slight abuse of notation $\alpha_{M+1} = 0$. Therefore $\exists \mathbf{G}^n \in \{-1, 0, 1\}^{M \times M}$ such that $g(\boldsymbol{\alpha}) = \mathbf{G}^n \boldsymbol{\alpha}$ for all $\boldsymbol{\alpha} \in \Delta_M^n$.

Combining observations 1,2&3, we can re-write the original optimization problem as follows:

$$\begin{aligned}
\min_{\boldsymbol{\alpha} \in \Delta_M} h(\boldsymbol{\alpha}) &= \min_{\boldsymbol{\alpha} \in \Delta_M^1 \cup ... \cup \Delta_M^{M!}} \boldsymbol{\gamma}^\top g(\boldsymbol{\alpha}) \\
&= \min_{n \in [M!]} \left\{ \min_{\boldsymbol{\alpha} \in \Delta_M^n} \boldsymbol{\gamma}^\top g(\boldsymbol{\alpha}) \right\} \\
&= \min_{n \in [M!]} \left\{ \min_{\boldsymbol{\alpha} \in \mathcal{H}(\mathcal{P}^n)} \boldsymbol{\gamma}^\top \mathbf{G}^n \boldsymbol{\alpha} \right\} \\
&\overset{(a)}{=} \min_{n \in [M!]} \left\{ \min_{\mathbf{p} \in \mathcal{P}^n} \boldsymbol{\gamma}^\top \mathbf{G}^n \mathbf{p} \right\} \\
&= \min_{n \in [M!], k \in [M]} \boldsymbol{\gamma}^\top g(\mathbf{p}_k^n)
\end{aligned} \tag{58}$$

where (a) holds because the minimum of a linear function over the convex hull of a set of points $\mathcal{X}$ is obtained at one of the points in $\mathcal{X}$.

Thus, to solve the original optimization problem, we only need to evaluate $M!$ linear functions with $M$ vectors each, and pick the one that achieves the smallest value. Finally, we will now show that the search space can be significantly reduced from $M! \times M$ to $M$ possible solutions.

Let $\Delta_M^n$ be an arbitrary subset of $\Delta_M$ whose associated sorted indices are $i_1^n, i_2^n, ..., i_M^n$, and $\mathcal{P}^n = \{\mathbf{p}_k^n\}_k$ are the associated extreme points. We first note that, $\forall k \in [M]$, $g(\mathbf{p}_k^n) = \begin{bmatrix} 0 \; ... \; 0 \; \frac{1}{k} \; 0 \; ... \; 0 \end{bmatrix}^\top$ with $j_k^n = \max\{i_1^n, ..., i_k^n\}$ is the non-zero index. Therefore, we have that $\forall n, k$:

$$h(\mathbf{p}_k^n) = \boldsymbol{\gamma}^\top g(\mathbf{p}_k^n) = \frac{\gamma_{j_k^n}}{k} \tag{59}$$

Equation (59) reveals that, amongst all vectors $\mathbf{p}_k^n$ with fixed $k$, the smallest error is always achieved by the subset $n$ whose associated $j_k^n$ index is the smallest, since the robust errors are always assumed to be sorted. Furthermore, the smallest value that $j_k^n$ can achieve is $k$, since it is the largest index amongst $k$ arbitrary indices from $[M]$. Therefore, let $\Delta_M^m$ be the subset whose sorting indices are $i_k = k$, i.e. $\boldsymbol{\alpha} \in \Delta_M^m$ implies $\alpha_1 \geq ... \geq \alpha_M$. For this subset, we will always have $j_k^m = \max\{1, ..., k\} = k$ which implies that $\forall n \in [M!]$ and $\forall k \in [M]$:

$$h(\mathbf{p}_k^n) = \frac{\gamma_{j_k^n}}{k} \geq \frac{\gamma_k}{k} = h(\mathbf{p}_k^m) \tag{60}$$

where $\mathbf{p}_k^m = \begin{bmatrix} \frac{1}{k} \; ... \; \frac{1}{k} \; 0 \; ... \; 0 \end{bmatrix}^\top$. Combining (58)&(60) we obtain:

$$\min_{\boldsymbol{\alpha} \in \Delta_M} h(\boldsymbol{\alpha}) = \min_{k \in [M]} \boldsymbol{\gamma}^\top g(\mathbf{p}_k^m) = \min_{k \in [M]} \frac{\gamma_k}{k} = \frac{\gamma_{k^*}}{k^*} \tag{61}$$

which can be achieved using $\boldsymbol{\alpha}^* = \begin{bmatrix} \frac{1}{k^*} \; ... \; \frac{1}{k^*} \; 0 \; ... \; 0 \end{bmatrix}^\top$. $\qquad\square$

### B.4.2 MAIN PROOF

We now restate and prove Theorem 2:

**Theorem** (Restated). *For a perturbation set $\mathcal{S}$, data distribution $\mathcal{D}$, and collection of $M$ classifiers $\mathcal{F}$ with individual adversarial risks $\eta_i$ ($i \in [M]$) such that $0 < \eta_1 \leq ... \leq \eta_M \leq 1$, we have $\forall \boldsymbol{\alpha} \in \Delta_M$:*

$$\min_{k \in [M]} \left\{ \frac{\eta_k}{k} \right\} \leq \eta(\boldsymbol{\alpha}) \leq \eta_M \tag{62}$$

*Both bounds are tight in the sense that if all that is known about the setup $\mathcal{F}$, $\mathcal{D}$, and $\mathcal{S}$ is $\{\eta_i\}_{i=1}^M$, then there exist no tighter bounds. Furthermore, the upper bound is always met if $\boldsymbol{\alpha} = \mathbf{e}_M$, and the lower bound (if achievable) can be met if $\boldsymbol{\alpha} = \begin{bmatrix} \frac{1}{m} \; ... \; \frac{1}{m} \; 0 \; ... \; 0 \end{bmatrix}^\top$, where $m = \arg\min_{k \in [M]} \{ \frac{\eta_k}{k} \}$.*

*Proof.* We first prove the upper bound and then the lower bound.

**Upper bound**: From Proposition 1, we have that $\eta$ is convex in $\boldsymbol{\alpha} \in \Delta_M$. Using $\Delta_M = \mathcal{H}(\{\mathbf{e}_1, ..., \mathbf{e}_M\})$ and applying Lemma 2, we get $\forall \boldsymbol{\alpha} \in \Delta_M$:

$$\eta(\boldsymbol{\alpha}) \leq \max_{\boldsymbol{\alpha} \in \Delta_M} \eta(\boldsymbol{\alpha}) = \max_{i \in [M]} \eta(\mathbf{e}_i) = \eta_M \tag{63}$$

This establishes the upper bound in (62). The bound is tight, since $\eta(\mathbf{e}_M) = \eta_M$ is achievable.

**Lower bound**: From Lemmas 4&5, we establish $\forall \boldsymbol{\alpha} \in \Delta_M$, the following result:

$$\eta(\boldsymbol{\alpha}) \geq \sum_{i=1}^M \left( (\eta_i - \eta_{i-1}) \cdot \max_{j \in \{i, ..., M\}} \{\alpha_j\} \right) = h(\boldsymbol{\alpha}) \geq \min_{\boldsymbol{\alpha} \in \Delta_M} h(\boldsymbol{\alpha}) = h(\boldsymbol{\alpha}^*) = \frac{\eta_m}{m} \tag{64}$$

where $m = \arg\min_{k \in [M]} \{ \frac{\eta_k}{k} \}$ and $\boldsymbol{\alpha}^* = \begin{bmatrix} \frac{1}{m} \; ... \; \frac{1}{m} \; 0 \; ... \; 0 \end{bmatrix}^\top$. This establishes the lower bound in (62).

The bound is tight, since for fixed $0 < \eta_1 \leq ... \leq \eta_M \leq 1$, we can construct $\mathcal{F}$, $\mathcal{S}$, and $\mathcal{D}$ such that $\eta(\boldsymbol{\alpha}) = h(\boldsymbol{\alpha})$ and $\forall i \in [M] : \eta(\mathbf{e}_i) = h(\mathbf{e}_i) = \eta_i$, as shown next.

Let $\mathcal{S} \subset \mathbb{R}^d$ be any closed and bounded set containing at least $M$ distinct vectors $\{\boldsymbol{\delta}_j\}_{j=1}^M \subseteq \mathcal{S}$. Let $\mathcal{D}$ be any valid distribution over $\mathcal{R} = \mathbb{R}^d \times [C]$ such that $\forall i \in [M]$: $\mathbb{P}\{z \in \mathcal{T}_i\} = \eta_i$, $\mathbb{P}\{z \in \mathcal{T}_{M+1}\} = 1$, and $\varnothing = \mathcal{T}_0 \subset \mathcal{T}_1 \subseteq \mathcal{T}_2 \subseteq ... \subseteq \mathcal{T}_M \subseteq \mathcal{T}_{M+1} \subset \mathcal{R}$. Finally, we construct classifiers $f_i$ $(\forall i \in [M])$ to satisfy the following assignment $\forall z \in \mathcal{T}_{M+1}$:

$$f_i(\mathbf{x} + \boldsymbol{\delta}) = y \ \ \forall \boldsymbol{\delta} \in \mathcal{S} \setminus \{\boldsymbol{\delta}_i\} \quad \& \quad f_i(\mathbf{x} + \boldsymbol{\delta}_i) = \begin{cases} y & \text{if } (\mathbf{x}, y) \notin \mathcal{T}_i \\ y' \neq y & \text{otherwise} \end{cases} \tag{65}$$

i.e., the $i$-th classifier decision $f_i(\mathbf{x} + \boldsymbol{\delta})$ is incorrect only if $\boldsymbol{\delta} = \boldsymbol{\delta}_i$ and $z \in \mathcal{T}_i$.

Given the above construction, we establish

$$
\begin{aligned}
\eta(\boldsymbol{\alpha}) &= \mathbb{E}_{z \sim \mathcal{D}} \left[ \max_{\boldsymbol{\delta} \in \mathcal{S}} \sum_{i=1}^M \alpha_i \mathbb{1}\{f_i(\mathbf{x} + \boldsymbol{\delta}) \neq y\} \right] \\
&\overset{(a)}{=} \int_{z \in \mathcal{T}_{M+1}} p_z(\mathbf{z}) \cdot \left( \max_{\boldsymbol{\delta} \in \mathcal{S}} \sum_{i=1}^M \alpha_i \mathbb{1}\{f_i(\mathbf{x} + \boldsymbol{\delta}) \neq y\} \right) d\mathbf{z} \\
&\overset{(b)}{=} \sum_{k=1}^M \int_{z \in \mathcal{T}_k \setminus \mathcal{T}_{k-1}} p_z(\mathbf{z}) \cdot \left( \max_{\boldsymbol{\delta} \in \mathcal{S}} \sum_{i=1}^M \alpha_i \mathbb{1}\{f_i(\mathbf{x} + \boldsymbol{\delta}) \neq y\} \right) d\mathbf{z} \\
&\overset{(c)}{=} \sum_{k=1}^M \int_{z \in \mathcal{T}_k \setminus \mathcal{T}_{k-1}} p_z(\mathbf{z}) \cdot \left( \max_{j \in \{k,..,M\}} \{\alpha_j\} \right) d\mathbf{z} \\
&= \sum_{k=1}^M \left[ \left( \max_{j \in \{k,..,M\}} \{\alpha_j\} \right) \int_{\mathcal{T}_k \setminus \mathcal{T}_{k-1}} p_z(\mathbf{z}) \, d\mathbf{z} \right] \\
&\overset{(d)}{=} \sum_{k=1}^M \left[ (\eta_i - \eta_{i-1}) \cdot \max_{j \in \{k,..,M\}} \{\alpha_j\} \right] = h(\boldsymbol{\alpha})
\end{aligned}
\tag{66}
$$

where: (a) holds because $\mathbb{P}\{z \in \mathcal{T}_{M+1}\} = 1$; (b) holds because we can partition $\mathcal{T}_{M+1}$ into $M+1$ sets: $\mathcal{T}_1 \cup (\mathcal{T}_2 \setminus \mathcal{T}_1) \cup ... \cup (\mathcal{T}_{M+1} \setminus \mathcal{T}_M)$, and because the max term is $0$ $\forall z \in \mathcal{T}_{M+1} \setminus \mathcal{T}_M$; (c) holds by construction of $\mathcal{F}$ and $\mathcal{S}$, and (d) holds since $\eta_i = \mathbb{P}\{z \in \mathcal{T}_i\}$ and $\mathcal{T}_i \subseteq \mathcal{T}_{i+1}$.

$\square$

### B.5 PROOF OF THEOREM 3

First, we state the classic result on the convergence of the projected sub-gradient method for convex minimization (Shor (2012)):

**Lemma 6** (Projected Sub-gradient Method). *Let $h : \mathbb{R}^d \to \mathbb{R}$ be a a convex and sub-differentiable function. Let $\mathcal{C} \subset \mathbb{R}^d$ be a convex set. For iterations $t = 1, .., T$, define the projected sub-gradient method:*

$$\mathbf{x}^{(t+1)} = \Pi_{\mathcal{C}} \left( \mathbf{x}^{(t)} - a_t \mathbf{g}^{(t)} \right) \tag{67}$$

$$h_{\text{best}}^{(t+1)} = \min \left\{ h_{\text{best}}^{(t)}, h(\mathbf{x}^{(t+1)}) \right\} \tag{68}$$

*where $a_t = a/t$ for some positive $a > 0$, $\mathbf{x}^{(1)} \in \mathcal{C}$ is an arbitrary initial guess, $h_{\text{best}}^{(1)} = h(\mathbf{x}^{(1)})$, and $\mathbf{g}^{(t)} \in \partial h(\mathbf{x}^{(t)})$ is a sub-gradient of $h$ at $\mathbf{x}^{(t)}$. Let $t_{\text{best}}$ designate the best iteration index thus far. Then, if $h$ has norm-bounded sub-gradients: $\|\mathbf{g}\|_2 \leq G$ for all $\mathbf{g} \in \partial h(\mathbf{x})$ and $\mathbf{x} \in \mathcal{C}$, we have:*

$$h_{\text{best}}^{(t)} - h^* \leq \frac{\|\mathbf{x}^{(1)} - \mathbf{x}^*\|_2^2 + G^2 \sum_{k=1}^t a_t^2}{2 \sum_{k=1}^t a_k} \xrightarrow[t \to \infty]{} 0 \tag{69}$$

*where:*

$$h^* = h(\mathbf{x}^*) = \min_{\mathbf{x} \in \mathcal{C}} h(\mathbf{x}) \tag{70}$$

We then prove Theorem 3 (restated below) via a direct application of Lemma 6:

**Theorem** (Restated). *The OSP algorithm output $\boldsymbol{\alpha}_T$ satisfies:*

$$0 \leq \hat{\eta}(\boldsymbol{\alpha}_T) - \hat{\eta}(\boldsymbol{\alpha}^*) \leq \frac{\|\boldsymbol{\alpha}^{(1)} - \boldsymbol{\alpha}^*\|_2^2 + Ma^2 \sum_{t=1}^{T} t^{-2}}{2a \sum_{t=1}^{T} t^{-1}} \xrightarrow[T \to \infty]{} 0 \tag{71}$$

*for any initial condition $\boldsymbol{\alpha}^{(1)} \in \Delta_M$, $a > 0$, where $\boldsymbol{\alpha}^*$ is a global minimum.*

*Proof.* The ensemble empirical adversarial risk $\hat{\eta}$ is convex and sub-differentiable (Proposition 1), being minimized over a convex set $\Delta_M$. At each iteration $t$ in OSP, the vector $\mathbf{g}$ obtained at line (12) is norm-bounded with $G = \sqrt{M}$, the vector $\mathbf{g}$ is also a sub-gradient of $\hat{\eta}$ at $\boldsymbol{\alpha}^{(t)}$, therefore Lemma 6 applies. $\qquad\square$

## C ADDITIONAL THEORETICAL RESULTS

### C.1 SPECIAL CASE OF THREE CLASSIFIERS

In this section, we derive a simplified search strategy for finding the optimal sampling probability for the special case of $M = 3$, akin to Section 4.2. Similar to (7), we can enumerate all possible classifiers/data-point configurations around $\mathcal{S}$, which allows us to write $\forall \boldsymbol{\alpha} \in \Delta_3$:

$$
\begin{aligned}
\eta(\boldsymbol{\alpha}) = {} & p_1 \cdot \max\{\alpha_1, \alpha_2, \alpha_3\} \\
& + p_2 \cdot \max\{\alpha_1 + \alpha_2, \alpha_3\} + p_3 \cdot \max\{\alpha_2 + \alpha_3, \alpha_1\} + p_4 \cdot \max\{\alpha_1 + \alpha_3, \alpha_2\} \\
& + p_5 \cdot \max\{\alpha_1, \alpha_2\} + p_6 \cdot \max\{\alpha_2, \alpha_3\} + p_7 \cdot \max\{\alpha_1, \alpha_3\} \\
& + p_8 \cdot \alpha_1 + p_9 \cdot \alpha_2 + p_{10} \cdot \alpha_3 + p_{11} \cdot 1 + p_{12} \cdot 0
\end{aligned}
\tag{72}
$$

where $\mathbf{p} \in \Delta_{12}$. Using (72), we obtain the following result:

**Theorem 4.** *Define $\mathcal{A} \subset \Delta_3$ to be the set of the following vectors:*

$$
\mathcal{A} = \left\{
\begin{bmatrix} 1 \\ 0 \\ 0 \end{bmatrix},
\begin{bmatrix} 0 \\ 1 \\ 0 \end{bmatrix},
\begin{bmatrix} 0 \\ 0 \\ 1 \end{bmatrix},
\begin{bmatrix} 1/2 \\ 1/2 \\ 0 \end{bmatrix},
\begin{bmatrix} 0 \\ 1/2 \\ 1/2 \end{bmatrix},
\begin{bmatrix} 1/2 \\ 0 \\ 1/2 \end{bmatrix},
\begin{bmatrix} 1/2 \\ 1/4 \\ 1/4 \end{bmatrix},
\begin{bmatrix} 1/4 \\ 1/2 \\ 1/4 \end{bmatrix},
\begin{bmatrix} 1/4 \\ 1/4 \\ 1/2 \end{bmatrix},
\begin{bmatrix} 1/3 \\ 1/3 \\ 1/3 \end{bmatrix}
\right\}
\tag{73}
$$

*Then for any three classifiers $f_1$, $f_2$, and $f_3$, perturbation set $\mathcal{S} \subset \mathbb{R}^d$, and data distribution $\mathcal{D}$, we have:*

$$
\min_{\boldsymbol{\alpha} \in \Delta_3} \eta(\boldsymbol{\alpha}) = \min_{\boldsymbol{\alpha} \in \mathcal{A}} \eta(\boldsymbol{\alpha})
\tag{74}
$$

*The set $\mathcal{A}$ is optimal, in the sense that there exist no smaller set $\mathcal{A}'$ such that (74) holds.*

Theorem 4 simplifies the search for the optimal sampling probability significantly, as it is sufficient to evaluate $\eta$ at exactly 10 different candidate solutions, captured by $\mathcal{A}$, and pick the best performing one. Theorem 4 also guarantees that the search procedure is efficient, since every candidate solution in $\mathcal{A}$ needs to be evaluated.

*Proof.* We shall use the same technique used in the proof of Lemma 5. We can decompose $\Delta_3$ into 6 such subsets $\Delta_3^1, ..., \Delta_3^6$, such that each subset contains vectors that share the same sorting indices. These subsets are convex, and they can be represented as the convex hull of three vectors. Due to the symmetry of the problem, we shall focus on one subset $\Delta_3^1 = \mathcal{H}\left(\left\{(1,0,0)^\top, (1/2, 1/2, 0)^\top, (1/3, 1/3, 1/3)^\top\right\}\right)$ where $\forall \boldsymbol{\alpha} \in \Delta_3^1$, we have: $\alpha_1 \geq \alpha_2 \geq \alpha_3$. Notice that for any $\boldsymbol{\alpha} \in \Delta_3^1$, all the terms in (72) become linear in $\boldsymbol{\alpha}$, except for the term $\max\{\alpha_2 + \alpha_3, \alpha_1\}$. Therefore, we can further decompose $\Delta_3^1$ into two convex subsets $\Delta_3^{1,1}$ and $\Delta_3^{1,2}$, such that:

$$
\Delta_3^{1,1} = \{\boldsymbol{\alpha} \in \Delta_3^1 : \alpha_1 \geq \alpha_2 + \alpha_3\} \quad \Delta_3^{1,2} = \{\boldsymbol{\alpha} \in \Delta_3^1 : \alpha_1 \leq \alpha_2 + \alpha_3\}
\tag{75}
$$

and $\eta$ is linear over both subsets (but not their union).

**Claim**: we have:

$$
\begin{aligned}
\Delta_3^{1,1} &= \mathcal{H}\left(\left\{(1,0,0)^\top, (1/2, 1/2, 0)^\top, (1/2, 1/4, 1/4)^\top\right\}\right) \\
\Delta_3^{1,2} &= \mathcal{H}\left(\left\{(1/3, 1/3, 1/3)^\top, (1/2, 1/2, 0)^\top, (1/2, 1/4, 1/4)^\top\right\}\right)
\end{aligned}
\tag{76}
$$

Since both $\Delta_3^{1,1}$ and $\Delta_3^{1,2}$ are convex, it is enough to show that:

$$
\begin{aligned}
\Delta_3^{1,1} &\subseteq \mathcal{H}\left(\left\{(1,0,0)^\top, (1/2, 1/2, 0)^\top, (1/2, 1/4, 1/4)^\top\right\}\right) \\
\Delta_3^{1,2} &\subseteq \mathcal{H}\left(\left\{(1/3, 1/3, 1/3)^\top, (1/2, 1/2, 0)^\top, (1/2, 1/4, 1/4)^\top\right\}\right)
\end{aligned}
\tag{77}
$$

for (76) to hold. For all $\boldsymbol{\alpha} \in \Delta_3^{1,1}$, define:

$$
\lambda_1 = \alpha_1 - \alpha_2 - \alpha_3 \geq 0, \quad \lambda_2 = 2 \cdot (\alpha_2 - \alpha_3) \geq 0, \ \& \ \lambda_3 = 4\alpha_3 \geq 0
\tag{78}
$$

Then we always have:

$$\boldsymbol{\alpha} = \lambda_1 \cdot \begin{bmatrix} 1 \\ 0 \\ 0 \end{bmatrix} + \lambda_2 \cdot \begin{bmatrix} 1/2 \\ 1/2 \\ 0 \end{bmatrix} + \lambda_1 \cdot \begin{bmatrix} 1/2 \\ 1/4 \\ 1/4 \end{bmatrix} \tag{79}$$

where it is easy to verify that $\lambda_1 + \lambda_2 + \lambda_3 = 1$. The same can be shown for any $\boldsymbol{\alpha} \in \Delta_3^{1,2}$, using the following:

$$\lambda_1 = 2 \cdot (\alpha_2 - \alpha_3) \geq 0, \quad \lambda_2 = 4 \cdot (\alpha_1 - \alpha_2) \geq 0, \ \& \ \lambda_3 = 3 \cdot (\alpha_2 + \alpha_3 - \alpha_1) \geq 0 \tag{80}$$

which establishes the claim in (76).

Using (76) and the linearity of $\eta$ on each subset, we can write:

$$
\begin{aligned}
\min_{\boldsymbol{\alpha} \in \Delta_3^1} \eta(\boldsymbol{\alpha}) &= \min \left\{ \min_{\boldsymbol{\alpha} \in \Delta_3^{1,1}} \eta(\boldsymbol{\alpha}), \min_{\boldsymbol{\alpha} \in \Delta_3^{1,2}} \eta(\boldsymbol{\alpha}) \right\} \\
&= \min \left\{ \eta \left( \begin{bmatrix} 1 \\ 0 \\ 0 \end{bmatrix} \right), \eta \left( \begin{bmatrix} 1/2 \\ 1/2 \\ 0 \end{bmatrix} \right), \eta \left( \begin{bmatrix} 1/2 \\ 1/4 \\ 1/4 \end{bmatrix} \right), \eta \left( \begin{bmatrix} 1/3 \\ 1/3 \\ 1/3 \end{bmatrix} \right) \right\}
\end{aligned} \tag{81}
$$

Finally, repeating this procedure for the remainder 5 sets $\Delta_3^2, ..., \Delta_3^6$ establishes (74). To show that the set $\mathcal{A}$ is minimal, we provide 10 constructions of $\eta$ using the $\mathbf{p}$ vector in (72) such that the $i^{\text{th}}$ vector $\boldsymbol{\alpha} \in \mathcal{A}$ is a unique (amongst $\mathcal{A}$) global optimum of $\eta$ characterized by the $i^{\text{th}}$ $\mathbf{p}$ vector (listed below):

$$
\begin{aligned}
\mathbf{p}_1 &= \begin{bmatrix} 0\ 0\ 0\ 0\ 0\ 0\ 0\ 0\ \dfrac{1}{2}\ \dfrac{1}{2}\ 0\ 0 \end{bmatrix}^\top \\[4pt]
\mathbf{p}_2 &= \begin{bmatrix} 0\ 0\ 0\ 0\ 0\ 0\ 0\ \dfrac{1}{2}\ 0\ \dfrac{1}{2}\ 0\ 0 \end{bmatrix}^\top \\[4pt]
\mathbf{p}_3 &= \begin{bmatrix} 0\ 0\ 0\ 0\ 0\ 0\ 0\ \dfrac{1}{2}\ \dfrac{1}{2}\ 0\ 0\ 0 \end{bmatrix}^\top \\[4pt]
\mathbf{p}_4 &= \begin{bmatrix} 0\ 0\ 0\ 0\ \dfrac{1}{2}\ 0\ 0\ 0\ 0\ \dfrac{1}{2}\ 0\ 0 \end{bmatrix}^\top \\[4pt]
\mathbf{p}_5 &= \begin{bmatrix} 0\ 0\ 0\ 0\ 0\ \dfrac{1}{2}\ 0\ \dfrac{1}{2}\ 0\ 0\ 0\ 0 \end{bmatrix}^\top \\[4pt]
\mathbf{p}_6 &= \begin{bmatrix} 0\ 0\ 0\ 0\ 0\ 0\ \dfrac{1}{2}\ 0\ \dfrac{1}{2}\ 0\ 0\ 0 \end{bmatrix}^\top \\[4pt]
\mathbf{p}_7 &= \begin{bmatrix} 0\ 0\ \dfrac{1}{2}\ 0\ 0\ \dfrac{1}{2}\ 0\ 0\ 0\ 0\ 0\ 0 \end{bmatrix}^\top \\[4pt]
\mathbf{p}_8 &= \begin{bmatrix} 0\ 0\ 0\ \dfrac{1}{2}\ 0\ 0\ \dfrac{1}{2}\ 0\ 0\ 0\ 0\ 0 \end{bmatrix}^\top \\[4pt]
\mathbf{p}_9 &= \begin{bmatrix} 0\ \dfrac{1}{2}\ 0\ 0\ \dfrac{1}{2}\ 0\ 0\ 0\ 0\ 0\ 0\ 0 \end{bmatrix}^\top \\[4pt]
\mathbf{p}_{10} &= \begin{bmatrix} 1\ 0\ 0\ 0\ 0\ 0\ 0\ 0\ 0\ 0\ 0\ 0 \end{bmatrix}^\top
\end{aligned} \tag{82}
$$

$\square$

## C.2 Worst Case Performance of Deterministic Ensembles

In Section 4.3, we showed via Theorem 2 that the adversarial risk of any randomized ensemble classifier is upper bounded by the worst performing classifier in the ensemble $\mathcal{F}$. In this section, we will show that the same cannot be said regarding deterministic ensemble classifiers. That is, there exist an ensemble $\mathcal{F}$, data distribution $\mathcal{D}$, and perturbation set $\mathcal{S}$ such that:

$$\eta(\bar{f}) > \max_{i \in [M]} \eta(f_i) \tag{83}$$

where $\bar{f}$ is the deterministic ensemble classifier constructed via the rule:

$$\bar{f}(\mathbf{x}) = \arg\max_{c \in [C]} \left[ \sum_{i=1}^{M} \tilde{f}_i(\mathbf{x}) \right]_c \tag{84}$$

Consider the following setup:

1. two binary classifiers in $\mathbb{R}^2$:

$$f_i(\mathbf{x}) = \begin{cases} 1 & \text{if } \mathbf{w}_i^\top \mathbf{x} \geq 0 \\ 2 & \text{otherwise} \end{cases} \tag{85}$$

which can be obtained from the "soft" classifiers:

$$\tilde{f}_i(\mathbf{x}) = \begin{bmatrix} \mathbf{w}_i^\top \mathbf{x} \\ -\mathbf{w}_i^\top \mathbf{x} \end{bmatrix} \tag{86}$$

using $f_i(\mathbf{x}) = \arg\max_{c \in \{1,2\}} [\tilde{f}_i(\mathbf{x})]_c$, where $\mathbf{w}_1 = [1 \ 1]^\top$ and $\mathbf{w}_2 = [1 \ -1]^\top$.

2. a $\text{Ber}(p)$ data distribution $\mathcal{D}$ over two data-points in $\mathbb{R}^2 \times [2]$:

$$\mathbf{z}_1 = (\mathbf{x}_1, y_1) = \left( \begin{bmatrix} -1 \\ 2 \end{bmatrix}, 1 \right) \quad \text{and} \quad \mathbf{z}_2 = (\mathbf{x}_2, y_2) = \left( \begin{bmatrix} -1 \\ -2 \end{bmatrix}, 1 \right) \tag{87}$$

3. the $\ell_2$ norm-bounded perturbation set $\mathcal{S} = \{ \boldsymbol{\delta} : \|\boldsymbol{\delta}\| \leq \epsilon \}$ for some $0 < \epsilon < 1/\sqrt{2}$.

We first note that for binary linear classifiers and $\ell_2$-norm bounded adversaries, we have that:

- the shortest distance between a point $\mathbf{x}$ and the decision boundary of linear classifier $f$ with weight $\mathbf{w}$ and bias $b$ is:

$$\zeta = \frac{|\mathbf{w}^\top \mathbf{x} + b|}{\|\mathbf{w}\|} \tag{88}$$

- if $f(\mathbf{x}) \neq y$, then the optimal adversarial perturbation is given by:

$$\boldsymbol{\delta} = -\text{sign}\left( \mathbf{w}^\top \mathbf{x} + b \right) \frac{\epsilon \mathbf{w}}{\|\mathbf{w}\|} \tag{89}$$

We can now evaluate the adversarial risks of each classifier:

$$\begin{aligned} \eta_1 &= p \cdot \left( \max_{\|\boldsymbol{\delta}\| \leq \epsilon} \mathbb{1} \left\{ \mathbf{w}_1^\top (\mathbf{x}_1 + \boldsymbol{\delta}) < 0 \right\} \right) + (1-p) \cdot \left( \max_{\|\boldsymbol{\delta}\| \leq \epsilon} \mathbb{1} \left\{ \mathbf{w}_1^\top (\mathbf{x}_2 + \boldsymbol{\delta}) < 0 \right\} \right) \\ &= p \cdot \left( \mathbb{1} \left\{ 1 - \sqrt{2}\epsilon < 0 \right\} \right) + (1-p) \cdot \left( \mathbb{1} \left\{ -3 < 0 \right\} \right) = 1 - p \end{aligned} \tag{90}$$

where we use $\epsilon < 1/\sqrt{2}$. Due to symmetry, we also get $\eta_2 = p$.

The average ensemble classifier $\bar{f}$ constructed from $f_1$ and $f_2$ is defined via the rule:

$$\bar{f}(\mathbf{x}) = \begin{cases} 1 & \text{if } x_1 \geq 0 \\ 2 & \text{otherwise} \end{cases} \tag{91}$$

whose adversarial risk can be computed as follows:

$$\begin{aligned} \bar{\eta} &= p \cdot \left( \max_{\|\boldsymbol{\delta}\| \leq \epsilon} \mathbb{1} \left\{ x_{1,1} + \delta_1 \right\} < 0 \right) + (1-p) \cdot \left( \max_{\|\boldsymbol{\delta}\| \leq \epsilon} \mathbb{1} \left\{ x_{2,1} + \delta_1 \right\} < 0 < 0 \right) \\ &= p \cdot \left( \mathbb{1} \left\{ -1 < 0 \right\} \right) + (1-p) \cdot \left( \mathbb{1} \left\{ -1 < 0 \right\} \right) = p + 1 - p = 1 \end{aligned} \tag{92}$$

which is strictly greater than $\max\{p, 1-p\} \ \forall p \in (0, 1)$. Therefore, we have constructed an example where deterministic ensembling is always worse than using any of the individual classifiers, which proves that deterministic ensemble classifiers *do not* satisfy the upper bound.

# D ADDITIONAL EXPERIMENTS AND COMPARISONS

## D.1 EXPERIMENTAL SETUP

In this section, we describe the complete experimental setup used for all our experiments.

**Training**. All models are trained for 100 epochs via SGD with a batch size of 256 and 0.1 initial learning rate, decayed by 0.1 first at the $50^{\text{th}}$ epoch and twice at the $75^{\text{th}}$ epoch. We employ the recently proposed margin-maximizing cross-entropy (MCE) loss from Zhang et al. (2022) with 0.9 momentum and a weight decay factor of $5 \times 10^{-4}$. We use 10 attack iterations during training with $\epsilon = 8/255$ and a step size $\beta = 2/255$. For IAT, each classifier is indepdenelty trained from a different random initialization, using a standard PGD adversary. For MRBoost, we use their public implementation from GitHub to reproduce all their results. For BARRE, we use an adaptive PGD (APGD) adversary (discussed in detail in Section D.4) as our training attack algorithm. We apply OSP for $T_o = 10$ iterations every $E_o = 10$ epochs.

To avoid catastrophic overfitting (Rice et al., 2020), we always save the best performing checkpoint during training. Since all the ensemble methods considered reduce to adversarial training for the first iteration, we use a shared adversarially trained first classifier. Doing so ensures a fair comparison between different ensemble methods. For both CIFAR-10, and CIFAR-100 datasets, we adopt standard data augmentation (random crops and flips). Per standard practice, we apply input normalization as part of the model, so that the adversary operates on physical images $\mathbf{x} \in [0, 1]^d$.

**Evaluation**. For all our robust evaluations, we will adopt the state-of-the-art ARC algorithm (Dbouk & Shanbhag, 2022) which can be used for both RECs and single models. Specifically, we use 20 iterations of ARC, with an attack strength $\epsilon = 8/255$ and approximation parameter $G = 2$. Following the recommendations of Dbouk & Shanbhag (2022), we use a step size of $2/255$ when evaluating single models ($M = 1$) and a step size of $8/255$ when evaluating RECs ($M \geq 2$).

## D.2 INDIVIDUAL MODEL ROBUSTNESS

In Tables 3&4, we provide the clean and robust accuracies of all the individual classifiers constructed via the different ensemble methods on CIFAR-10 and CIFAR-100, respectively. Robust accuracy is measured using ARC.

As expected, only ensembles produced via IAT consist of classifiers achieving near-identical robust and natural accuracies. In contrast, ensembles produced via MRBosst or BARRE witness a degradation in individual classifier robust accuracy as the ensemble size grows. However, since MRBoost was not initially designed for randomized ensemble classifiers, this degradation in robust accuracy can be rather severe as seen for MobileNetV1 in both Tables 3&4. This explains why, for such ensembles, the optimal sampling probability obtained for the constructed REC completely disregards the last classifier as highlighted in Section 5.1.

Table 3: Natural and robust accuracies of the individual classifiers of all ensembles methods trained on CIFAR-10 (from Table 2). Robust accuracy is measured against an $\ell_\infty$ norm-bounded adversary using ARC with $\epsilon = 8/255$.

| Network | Method | $f_1$ | | $f_2$ | | $f_3$ | | $f_4$ | | $f_5$ | |
|---|---|---|---|---|---|---|---|---|---|---|---|
| | | $A_{\text{nat}}$ | $A_{\text{rob}}$ | $A_{\text{nat}}$ | $A_{\text{rob}}$ | $A_{\text{nat}}$ | $A_{\text{rob}}$ | $A_{\text{nat}}$ | $A_{\text{rob}}$ | $A_{\text{nat}}$ | $A_{\text{rob}}$ |
| | IAT | | | 73.42 | 41.94 | 74.44 | 42.25 | 74.27 | 42.06 | 74.17 | 42.14 |
| ResNet-20 | MRBoost | 73.18 | 41.99 | 76.00 | 41.42 | 76.59 | 39.60 | 77.25 | 38.38 | 76.43 | 36.62 |
| | BARRE | | | 76.08 | 41.18 | 77.40 | 39.87 | 77.12 | 39.07 | 77.60 | 37.01 |
| | IAT | | | 79.17 | 46.21 | 79.05 | 46.60 | 78.44 | 46.11 | 78.76 | 46.74 |
| MobileNetV1 | MRBoost | 79.01 | 46.22 | 80.11 | 44.52 | 77.54 | 42.03 | 77.94 | 39.36 | 68.89 | 33.40 |
| | BARRE | | | 80.15 | 44.56 | 79.43 | 42.67 | 79.56 | 39.65 | 79.60 | 38.28 |
| | IAT | | | 80.64 | 48.23 | 81.24 | 48.83 | 81.13 | 48.70 | — | — |
| ResNet-18 | MRBoost | 80.96 | 48.72 | 84.01 | 47.56 | 83.67 | 45.72 | 83.88 | 43.38 | — | — |
| | BARRE | | | 84.35 | 46.48 | 84.89 | 45.86 | 83.88 | 43.09 | — | — |

Table 4: Natural and robust accuracies of the individual classifiers of all ensembles methods trained on CIFAR-100 (from Table 2). Robust accuracy is measured against an $\ell_\infty$ norm-bounded adversary using ARC with $\epsilon = 8/255$.

| NETWORK | METHOD | $f_1$ | | $f_2$ | | $f_3$ | | $f_4$ | | $f_5$ | |
|---|---|---|---|---|---|---|---|---|---|---|---|
| | | $A_{\text{nat}}$ | $A_{\text{rob}}$ | $A_{\text{nat}}$ | $A_{\text{rob}}$ | $A_{\text{nat}}$ | $A_{\text{rob}}$ | $A_{\text{nat}}$ | $A_{\text{rob}}$ | $A_{\text{nat}}$ | $A_{\text{rob}}$ |
| RESNET-20 | IAT | | | 38.64 | 17.68 | 38.40 | 17.89 | 39.13 | 17.63 | 38.36 | 18.13 |
| | MRBOOST | 38.34 | 17.69 | 41.69 | 17.29 | 42.69 | 17.67 | 42.92 | 17.44 | 42.83 | 16.11 |
| | BARRE | | | 41.57 | 18.22 | 42.96 | 17.24 | 42.69 | 17.14 | 43.72 | 16.30 |
| MOBILENETV1 | IAT | | | 51.46 | 23.01 | 50.61 | 23.00 | 51.21 | 23.40 | 51.89 | 23.56 |
| | MRBOOST | 51.87 | 23.45 | 53.96 | 22.63 | 53.45 | 20.48 | 52.55 | 19.90 | 38.88 | 11.34 |
| | BARRE | | | 52.75 | 22.90 | 53.61 | 21.21 | 54.31 | 18.67 | 51.99 | 18.02 |
| RESNET-18 | IAT | | | 53.85 | 24.17 | 54.80 | 24.30 | 54.71 | 24.50 | – | – |
| | MRBOOST | 53.85 | 24.15 | 54.78 | 22.28 | 47.49 | 16.28 | 48.13 | 15.98 | – | – |
| | BARRE | | | 55.21 | 22.26 | 55.69 | 21.05 | 53.73 | 17.99 | – | – |

## D.3 ATTACKS FOR RANDOMIZED ENSEMBLES

Given a data-point $z = (x, y)$ and a potentially random classifier $f$, the goal of an adversary is to find an adversarial perturbation that maximizes the single-point expected adversarial risk:

$$\delta^* = \arg\max_{\delta:\|\delta\|_p \leq \epsilon} r(z, \delta) = \arg\max_{\delta:\|\delta\|_p \leq \epsilon} \mathbb{E}_f\left[\mathbb{1}\left\{f(x + \delta) \neq y\right\}\right] = \arg\max_{\delta:\|\delta\|_p \leq \epsilon} \mathbb{P}\left\{f(x + \delta) \neq y\right\} \quad (93)$$

where we adopt the $\ell_p$ norm-bounded adversary for the remainder of this section.

Projected gradient descent (PGD) (Madry et al., 2018) is perhaps the most popular attack algorithm for solving (93) for the case of differentiable deterministic classifiers. Specifically, given a surrogate loss function $l$, such as the cross-entropy loss, PGD finds an adversarial $\delta$ iteratively via the following:

$$\delta^{(k)} = \Pi_\epsilon^p\left(\delta^{(k-1)} + \eta\mu_p\left(\nabla_x l\left(\tilde{f}\left(x + \delta^{(k-1)}\right), y\right)\right)\right) \quad (94)$$

where $\mu_p$ is the $\ell_p$ steepest direction projection operator, and $\Pi_\epsilon^p$ is the projection operator on the $\ell_p$ ball of radius $\epsilon$.

In order to adapt PGD for evaluating randomized ensemble classifiers, Pinot et al. (2020) first proposed adaptive PGD (APGD-L) using the expectation-over-transformation (EOT) method (Athalye

Table 5: Comparing the success of different attack algorithms at fooling various RECs using $\ell_\infty$ norm-bounded attacks with $\epsilon = 8/255$ on CIFAR-10. All the RECs are constructed with equiprobable sampling.

| Network | Method | APGD-L | APGD-S | ARC | ARC-R |
|---|---|---|---|---|---|
| ResNet-20 | IAT | 49.31 | 49.34 | 46.73 | **45.77** |
| | MRBoost | 49.65 | 49.61 | 47.74 | **46.66** |
| | BARRE | 49.79 | 49.75 | 48.05 | **47.35** |
| MobileNetV1 | IAT | 52.94 | 52.91 | 50.68 | **49.57** |
| | MRBoost | 51.19 | 51.02 | 49.37 | **48.05** |
| | BARRE | 52.16 | 51.94 | 51.16 | **49.91** |
| ResNet-18 | IAT | 54.50 | 54.49 | 52.42 | **51.43** |
| | MRBoost | 54.51 | 54.23 | 53.19 | **51.82** |
| | BARRE | 54.52 | 54.07 | 53.62 | **52.13** |

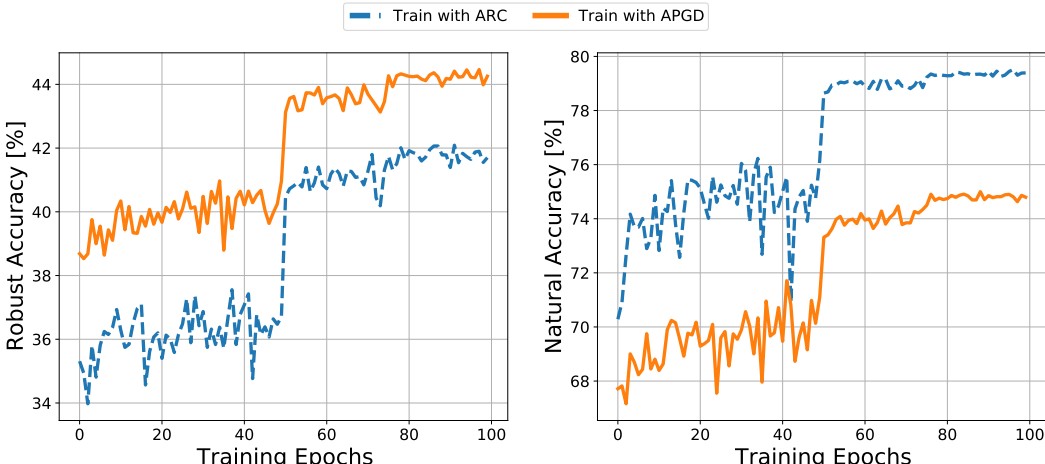

Figure 4: The robust (left) and natural (right) accuracies of an REC of two ResNet-20's trained on CIFAR-10 using BARRE vs. the training epochs of the second classifier $f_2$, where the first classifier $f_1$ is pre-adversarially trained. Robust and natural accuracies are reported on the test set, using $\ell_\infty$ norm-bounded adversaries with ARC and $\epsilon = 8/255$.

et al., 2018), which uses (94) with the expected loss function as follows:

$$\boldsymbol{\delta}^{(k)} = \Pi_\epsilon^p \left( \boldsymbol{\delta}^{(k-1)} + \eta\mu_p \left( \nabla_{\mathbf{x}}\mathbb{E}\left[ l\left( \tilde{f}\left( \mathbf{x} + \boldsymbol{\delta}^{(k-1)} \right), y \right) \right] \right) \right) \qquad (95)$$

Note that the discrete nature of randomized ensembles allows for an exact computation of the expectation in (95).

Recently, Zhang et al. (2022) proposed a stronger version of adaptive PGD, where the expectation is taken at the softmax level (APGD-S). Using APGD-S, Zhang et al. (2022) were able to compromise the BAT defense. Independently, Dbouk & Shanbhag (2022) studied the effectiveness of EOT-based adaptive attacks for evaluating the robustness of RECs, and concluded that such methods are fundamentally ill-suited for the task. Instead, they proposed the ARC attack (Algorithm 2 of (Dbouk & Shanbhag, 2022)), which relied on iteratively updating the perturbation based on estimating the direction towards the decision boundary of each classifier and using an adaptive step size method.

In this section, we propose a small modification to ARC (ARC-R) that proves to be quite more effective in the equiprobable setting. Specifically, instead of looping over the classifiers in a deterministic fashion based on the order of the sampling probability vector, we propose using a randomized order loop. This ensures that ARC is never biased towards certain classifiers. In fact, Table 5 demonstrates that ARC-R is better than APGD-L (Pinot et al., 2020), APGD-S Zhang et al. (2022), and ARC (Dbouk & Shanbhag, 2022) at evaluating the robustness of RECs on CIFAR-10, constructed with equiprobable sampling across various network architectures and ensemble training methods. Hence, we shall adopt this version of ARC for all our experiments.

### D.4 ARC VS. ADAPTIVE PGD FOR BARRE

As highlighted in Section 5.1, we find that ARC, despite being the strongest adversary, leads to poor performance when adopted as the training attack in BARRE. In this section, we investigate this phenomenon, as we study the performance of BARRE using two different attacks during training, APGD (Zhang et al., 2022) and ARC (Dbouk & Shanbhag, 2022). Specifically, we train two RECs on CIFAR-10 using the ResNet-20 architecture. Both RECs share the same first classifier $f_1$, which is adversarially trained using standard PGD. The second classifier $f_2$ is trained via either APGD or ARC.

Figure 4 plots the evolution of both robust and clean accuracies of the two RECs across the 100 training epochs of $f_2$, measured on the test set. Note that in both RECs, the robust accuracy is evaluated via the stronger ARC adversary. When evaluated on clean images, we find that BARRE with

ARC leads to significantly more accurate RECs when compared to BARRE with APGD. However, this comes at the expense of robust accuracy, as the REC obtained via BARRE with ARC is much more vulnerable than the APGD counterpart. We hypothesize that the adversarial samples generated via ARC during training do not generalize well to the test set. This explains why we observe that the REC obtained via BARRE with ARC achieves much higher robust accuracies on the *training* set. Thus, for better generalization performance, we shall adopt adaptive PGD during training in all our experiments.

### D.5 ADDITIONAL RESULTS

In this section, we complete the CIFAR-10 results reported in Table 1 for showcasing the benefit of randomization. Specifically, Table 6 provides further evidence that BARRE can train RECs of competitive robustness compared to MRBoost-trained deterministic ensembles, while requiring significantly less compute.

Table 6: Comparison between BARRE and MRBoost across different network architectures and ensemble sizes on CIFAR-100. Robust accuracy is measured against an $\ell_\infty$ norm-bounded adversary using ARC with $\epsilon = 8/255$.

| Network | Method | $M=1$ | | | $M=2$ | | | $M=3$ | | | $M=4$ | | |
| | | $A_\text{nat}$ | $A_\text{rob}$ | FLOPs | $A_\text{nat}$ | $A_\text{rob}$ | FLOPs | $A_\text{nat}$ | $A_\text{rob}$ | FLOPs | $A_\text{nat}$ | $A_\text{rob}$ | FLOPs |
| ResNet-20 | MRBoost | 38.34 | 17.69 | 81 M | 41.08 | 19.38 | 162 M | 42.60 | 20.48 | 243 M | 43.62 | 21.36 | 324 M |
| | BARRE | | | | 39.95 | 19.13 | 81 M | 40.96 | 19.85 | 81 M | 41.40 | 21.41 | 81 M |
| MobileNetV1 | MRBoost | 51.87 | 23.45 | 312 M | 54.41 | 25.73 | 624 M | 54.91 | 26.63 | 936 M | 55.03 | 26.97 | 1.2 B |
| | BARRE | | | | 52.31 | 24.96 | 312 M | 52.74 | 25.75 | 312 M | 53.14 | 27.12 | 312 M |
| ResNet-18 | MRBoost | 53.85 | 24.15 | 1.1 B | 55.83 | 25.99 | 2.2 B | 55.39 | 26.09 | 3.3 B | 55.80 | 26.50 | 4.4 B |
| | BARRE | | | | 54.53 | 25.37 | 1.1 B | 54.92 | 25.76 | 1.1 B | 54.63 | 26.90 | 1.1 B |

