# OpenReview forum: "On the Robustness of Randomized Ensembles to Adversarial Perturbations"
_ICLR.cc/2023/Conference — Submitted to ICLR 2023_

### Official Review · Reviewer_tU9P · 2022-10-16

**Confidence:** 4
**Correctness:** 4
**Technical Novelty And Significance:** 3
**Empirical Novelty And Significance:** 2
**Recommendation:** 6

**Clarity, Quality, Novelty And Reproducibility:**

The paper is well written. However, I still have some doubts on the reproducibility of experiments and on the novelty (see the Strength and Weaknesses Section for detailed comments)

**Strength And Weaknesses:**

Strength:

-Paper is well written

-The authors formally show that picking the optimal sampling probabilities for different classifiers is equivalent to optimising a piece-wise linear convex function over a convex set. This enables them to use tools from convex optimisation to solve this problem. I find this observation and the resulting algorithms interesting.


Weaknesses:

-Experimental results should be improved/clarified:

a) Table 1 and 2 report different trends in the comparison with MRBoost. In fact, while in table 1 MRBoost almost always obtains better accuracy and robustness compared to BARRE (the method presented by the authors), the trend is reversed in Table 2. This is particularly confusing for the ResNEt-18 M=4 example, where while BARRE has same values in both Table 1 and 2, for MRBoost these are different.

b) Please, report computational times. Also, it would be interesting to also consider larger values of M.

c) FLOPs in Table 1 is not defined


-Theorem 1 guarantees that to improve the empirical risk of the ensemble compared to that of the individual classifiers, region $\mathcal{R}_1$ should contain large probability mass wrt the input distribution. However, a discussion on how often this condition is verified in practice is missing. Intuitively, I believe that, due to the transferability of adversarial examples, such region will not be large in general. Nevertheless, a possibly interesting direction could be to train the individual classifiers in the ensemble with different adversarial training techniques in order to enforce that $\mathcal{R}_1$ is large, while keeping each classifier accurate and robust to a particular class of attacks.

-The fact that randomized classifiers can improve the adversarial robustness wrt the individual classifiers is well known. Therefore, also considering the above comments on the experiments, the results of the paper feel a bit incremental.

-Finally, the comparison with the literature completely misses Bayesian ensembles, which have been already shown to be robust to adversarial examples, see e.g. [1].


[1]: Carbone, Ginevra, et al. "Robustness of bayesian neural networks to gradient-based attacks." Advances in Neural Information Processing Systems 33 (2020): 15602-15613.


**Summary Of The Paper:**

The paper considers randomised ensemble classifiers (RECs) and study the problem of how to assign the probabilities to select each classifier to improve the adversarial robustness of the REC. The authors first show that minimising the adversarial risk is equivalent to minimising a piecewise linear function with linear constraints. Then, by relying on the convexity of the problem, they develop algorithms to solve the resulting optimisation problem first in the case of 2 classifiers and then they consider the general case.  Experimental analysis on CIFAR-10 and CIFAR-100 compare the proposed method with state-of-the-art approaches.

**Summary Of The Review:**

The paper studies an important problem and presents an interesting new perspective that makes possible to use of tools from convex optimisation. However, experimental results do not seem to present a substantial improvement compared to state-of-the-art.

---

> ### Author Response · Authors · 2022-11-15
> **Response to Reviewer tU9P (Part 1/2)**
>
> Thank you for the positive review. We hope our responses below address your concerns.
>
> “**Table 1 and 2 report different trends in the comparison with MRBoost. In fact, while in table 1 MRBoost almost always obtains better accuracy and robustness compared to BARRE (the method presented by the authors), the trend is reversed in Table 2. This is particularly confusing for the ResNEt-18 $M=4$ example, where while BARRE has same values in both Table 1 and 2, for MRBoost these are different.** ”
>
> MRBoost in Table 1 reports the performance of the *deterministic* ensemble classifier constructed from MRBoost trained ensemble $f_1$, …, $f_M$, whereas MRBoost in Table 2 reports the performance of the *randomized* ensemble classifier constructed from the same MRBoost trained ensemble $f_1$, …, $f_M$. This explains why the numbers and trends look different. The purpose of Table 1 is to show the benefit of randomization, whereas the purpose of Table 2 is to find the best training method for RECs.
>
> “**FLOPs in Table 1 is not defined** ”
>
> FLOPs is the number of floating-point operations required by each classifier for inference. We will clarify this in the paper.
>
> “**Please, report computational times** ”
>
> The computational time is directly proportional to FLOPs, which is why we felt reporting FLOPs would be sufficient.
>
> “**it would be interesting to also consider larger values of $M$.** ”
>
> We agree that larger $M$ (we have shown for $M\leq 5$) would be interesting, unfortunately we are limited with practical computational bottlenecks, as boosting methods in general tend to have long training times that scale poorly with $M$. This is why past works also restrict the values of $M$ to 2 (BAT) and $5$ (MRBoost).
>
> "**Theorem 1 guarantees that to improve the empirical risk of the ensemble compared to that of the individual classifiers, region $R_1$ should contain large probability mass wrt the input distribution. However, a discussion on how often this condition is verified in practice is missing. Intuitively, I believe that, due to the transferability of adversarial examples, such region will not be large in general.** "
>
> This is an interesting point. You are right that in practice, it is very difficult to make $Pr(R_1)$ very large (compared to $\eta_1$ and $\eta_2$). We can see this by calculating $Pr(R_1)$ using Theorem 1 which states that the optimal REC adversarial risk would be $\eta(\alpha^*) = 0.5(\eta_1+\eta_2-Pr(R_1))$ (assuming (8) is met). Using the Table 1 results, and the knowledge of the individual classifier performances (Table 3 in App C.2), one can calculate the empirical probability of region $R_1$ in the test set for each REC. For instance, for BARRE trained ResNet-20 on CIFAR-10, $Pr(R_1)$ is roughly 0.056 ($\eta_1 = 0.5801$, $\eta_2=0.5882$). As you pointed out, $Pr(R_1)$ is small compared to $\eta_1$ or $\eta_2$. We will mention this in the paper.

---

> > ### Author Response · Authors · 2022-11-15
> > **Response to Reviewer tU9P (Part 2/2)**
> >
> > "**Nevertheless, a possibly interesting direction could be to train the individual classifiers in the ensemble with different adversarial training techniques in order to enforce that R1 is large, while keeping each classifier accurate and robust to a particular class of attacks.**"
> >
> > In fact, BARRE already strives to maximize $Pr(R_1)$ while maintaining the robustness of f_2 and f_1 (minimize $\eta_1$ and $\eta_2$), in order to minimize the REC adversarial risk $\eta(\alpha^*) = 0.5(\eta_1+\eta_2-Pr(R_1))$ (from Theorem 1).
> > This is why BARRE initially adversarially trains a robust classifier $f_1$ (minimizing $\eta_1$), then trains $f_2$ (initialized from $f_1$ to minimizes $eta_2$) on the adversarial examples of the REC of $f_1$ and $f_2$ (constructed with equiprobable sampling (due to Theorem 1)). In this way, $\eta(\alpha^*)$ is maximized because if $Pr(R_1)$ is low (compared to $\eta_1$ and $\eta_2$) then a large portion (over the training set) of the generated adversarial examples will fool both $f_1$ and $f_2$, and thus $f_2$ will be trained to be robust to these perturbations, hence increasing $Pr(R_1)$ while maintaining $eta_2$ as low as possible.
> >
> > Note that the goal of our work is to lay the foundations for robust randomized ensemble classifiers, and show that one can construct them in practice. It would be interesting to see how one can improve BARRE to achieve better results in practice.
> >
> > "**The fact that randomized classifiers can improve the adversarial robustness wrt the individual classifiers is well known. Therefore, also considering the above comments on the experiments, the results of the paper feel a bit incremental.** "
> >
> > While it seems ‘intuitive’ that randomization should help with robustness, it can be deceptive. Past work has lacked a deep understanding regarding the adversarial vulnerability of randomized ensembles which can be seen in the ease with which the state-of-the-art method (BAT) was broken by independent researchers (MRBoost [Zhang et al., 2022] and ARC [Dbouk & Shanbhag, 2022]). In fact, ARC shows that a BAT-trained REC performs worse than a single classifier trained using standard AT. Our work provides a deeper understanding of the fundamental limits on the robustness achieved by RECs.
> >
> > "**Finally, the comparison with the literature completely misses Bayesian ensembles, which have been already shown to be robust to adversarial examples, see e.g. [1]** "
> >
> > Note that Bayesian NNs are fundamentally different from RECs. BNNs sample weights from a posterior distribution, and require multiple inferences with different sampled weights to perform one inference. In contrast, RECs sample one classifier at test time for inference. Nevertheless, we will modify the related works section to discuss BNNs.
> >
> > Finally, we hope that we have addressed your questions/concerns, and hope that you increase your score.

---

> ### Author Response · Authors · 2022-12-07
> **Waiting for Reviewer Feedback**
>
> Dear Reviewer tU9P,
>
> Thank you again for the positive review in addition to your comments and suggestions. As the discussion period soon comes to an end, we are looking forward to your feedback to our response and revised manuscript.
>
> We hope that we have addressed your concerns, and kindly ask that you revise your score.

---

### Official Review · Reviewer_3zJ9 · 2022-10-23

**Confidence:** 4
**Correctness:** 4
**Technical Novelty And Significance:** 3
**Empirical Novelty And Significance:** 3
**Recommendation:** 8

**Clarity, Quality, Novelty And Reproducibility:**

Clarity: the paper is highly technical but it can be understood as the definitions/results are presented with care, and some particular cases are studied in detail first before presenting the more general results. Overall the paper makes pretty clear statements.

Quality: the technical quality is high. Some lemmas/theorems have easier proofs than others but overall I think the results are interesting and non-trivial. Some proofs could be simplified as I mentioned. However, the quality suffers in the experimental section due to the absence of confidence intervals. When there is some randomness in the algorithms this is a must, in order to be able to *bold* the results and claim that there is improvement over the baseline.

Novelty: this results seem natural in the context of clean accuracy, and I would say it is possible that similar questions have been studied before. Even though I cannot say the results are not novel, as I have stated before the authors should do a better job of finding and discussing some relevant results.

Reproducibility: the authors have provided the code for the experiments

**Strength And Weaknesses:**

The main strength of this paper are the interesting theoretical bounds on the performance of randomized classifiers. This is important as the paradigm of training a single robust classifier appears to have hit a limit in terms of robustness, and the obtained lower bounds from Theorem 2 indicate that randomizing the classifier could lead to potentially big improvements. Of course, the lower bounds are only achieved when the classifiers at hand have the nice "incoherence" property where they don't share "vulnerable regions" where all are simultaneously mistaken by an adversary. The paper does a good job of presenting the case of two classifiers where the results are pretty straightforward and easy to understand, before presenting Theorem 2, which is the statement in full generality.

The math appears to be correct (I have checked some of the results in detail) and all the statements are clear, as well as the flow of the proofs is well organised. This is in contrast with the awful quality of the maths/proofs from average ICML/NeurIPS/ICLR submission, so this deserves a special mention. I would say however that some proofs are a bit more difficult than needed, for example in the proof of Lemma 5 in page 18, the items (1) and (2) are trivial. (1) simply follows from the fact that the sets are intersection of half-spaces and (2) follows from the well known equivalence between Vertex and Hyperplane descriptions of convex polytopes, see for example https://link.springer.com/content/pdf/bbm:978-0-387-46112-0/1.pdf

The theoretical results are not left without application, as they are used to define the algorithms 1 and 2, which builds a randomized classifier in stages, by adding robust classifiers while simultaneouly optimizing the sampling probabilities of the ensemble. Overall the theory justifies the design of the algorithm which is ideal.

The main weakness I see (unfortunately) is the empirical evaluation. Now, I think the settings studied are extensive, but the fact that there is randomness involved in the algorithm (through the use of SGD) means that one should provide some assessment of the confidence about the numbers stated in Table 1 and 2. The authors only provide a single number and it might be the case that the improvements are due to randomness. Hopefully this can be resolved simply by running multiple seeds and computing some confidence intervals. That would greatly improve the paper.

Finally another weakness is that the authors do not make an effort of presenting related results. I believe this topic should have been studied before at least in the context of clean accuracy (no adversary). The authors should comment on the relation to the papers:
1. https://aamas.csc.liv.ac.uk/Proceedings/aamas2014/aamas/p485.pdf
2. https://www.ncbi.nlm.nih.gov/pmc/articles/PMC3156487/
3. https://www.cs.waikato.ac.nz/~ml/publications/2002/bounds.pdf
4. https://ieeexplore.ieee.org/stamp/stamp.jsp?tp=&arnumber=7044723

and cite them, as it seems they study a similar question (possibly in the non-adversarial setting).

**Summary Of The Paper:**

The paper studies the limits of the adversarial robustness that can be obtained by combining $M$ different classifiers into one randomized classifier that at inference time, chooses one of the elements with certain probability. Some intermediate lower and upper bounds are presented that illustrate the main intuition behind the results. The main theorem is Theorem 2 which basically states that the randomized classifier in the worst case is as accurate as its worst component (trivial) and in the best case it can improve the robustness by a $1/M$ factor in the case where all classifiers have the same robustness (this is the interesting bound, and the result is stated in the more general case where each classifier might have a different robustness). Two algorithms are provided: the first obtains the best sampling probaility vector (assuming an oracle that obtains optimal adversarial perturbations) and the second is a boosting algorithm to obtain good classifiers to create the ensemble. In practical experiments it is shown that the second algorithm can lead to increased robustness against the state-of-the art ARC attack which was shown to break previous randomized ensemble defenses.

**After Rebuttal**
The authors have addressed some of my concerns. Despite not addressing the missing confidence intervals in the experiments, I am inclined to increase my score because of the following:

1. The method studied is interesting and significant, as there hasn't been an improvement in robust accuracy metrics from the single model paradigm in the last years. I believe most improvements have been on the training speed department rather than the robust accuracy metric. As such, exploring alternative paradigms like the randomized ensemble classifier studied here is one promising way forward.
2. A good amount of contributions are theoretical, and they provide guidance on how to develop better algorithms for building RECs. As such, I think it is possible to give the authors a pass regarding the fact that the improvements over the baseline are small and lack the confidence intervals. Focusing only on the experimental part would dismiss the interesting theoretical results which could be used or improved upon by others. Of course, after reflecting on the results presented, they appear quite intuitive, however, i think it is a stretch to call them 'trivial' as other reviewers have dismissed. To support such a statement they should have provided a simplified proof or argument, which I believe is not the case.
3. I disagree that not enjoying certain properties of BNNs would be a reason to dismiss researching RECs. Both are methods with pros and cons, and the goal of this paper as I understood was to advance the algorithmic and theoretical framework of RECs, rather than claim they are superior to BNNs in all aspects.
4. There are some concerns that this method is not suitable for some applications (like certain medical tasks). However, applications of ML have vastly different specifications and its hard to develop a one-size-fits-all method.

In the end, raising my score will not help much as there is still a big disagreement with two other reviewers. It would be most helpful if they re-evaluate their score after the author's rebuttal, and express if some of their concerns have been addressed.

**Summary Of The Review:**

Upper and lower bounds for the robustness of ensemble classifiers are presented. The results are quite interesting and their presentation is clear despite being highly technical. The theory leads to the design of principled algorithms that are evaluated against strong baselines, showing promise for improvement. Overall the paper seems to make significant contributions. However two big issues remain: (1) lack of discussion regarding prior results in the context of (non-adversarial) accuracy of ensembles and (2) computation of confidence intervals for the numbers presented in the tables. These two last points prevent me from confidently recommending acceptance but I would increase my score if they are addressed.

---

> ### Author Response · Authors · 2022-11-15
> **Response to Reviewer 3zJ9**
>
> Thank you for the positive review. We hope our responses below address your concerns.
>
> “**The math appears to be correct (I have checked some of the results in detail) and all the statements are clear, as well as the flow of the proofs is well organised. This is in contrast with the awful quality of the maths/proofs from average ICML/NeurIPS/ICLR submission, so this deserves a special mention.** ”
>
> We did spend a lot of effort in simplifying the proofs and presentation. We are happy you noticed.
>
> “**I would say however that some proofs are a bit more difficult than needed, for example in the proof of Lemma 5 in page 18, the items (1) and (2) are trivial. (1) simply follows from the fact that the sets are intersection of half-spaces and (2) follows from the well known equivalence between Vertex and Hyperplane descriptions of convex polytopes, see for example https://link.springer.com/content/pdf/bbm:978-0-387-46112-0/1.pdf**”
>
> Thank you for pointing out this reference. We agree that item (1) is trivial, as the proof is very elementary. The construction proof of item (2) is actually required as we use the explicit vertex description later in the proof of lemma (5), specifically in equations (59) and (60).
>
> “**The main weakness I see (unfortunately) is the empirical evaluation. Now, I think the settings studied are extensive, but the fact that there is randomness involved in the algorithm (through the use of SGD) means that one should provide some assessment of the confidence about the numbers stated in Table 1 and 2. The authors only provide a single number and it might be the case that the improvements are due to randomness. Hopefully this can be resolved simply by running multiple seeds and computing some confidence intervals. That would greatly improve the paper.** ”
>
> We do agree that reporting error bars to account for randomness of the SGD-based training process would be better. Please note that obtaining error bars for this, i.e., repeating the entire training run multiple times, is very prohibitive due to the long training times suffered by boosting frameworks in general. None of the past works on ensembles such as BAT or MRBoost have shown error bars for this reason.
>
> “**Finally another weakness is that the authors do not make an effort of presenting related results. I believe this topic should have been studied before at least in the context of clean accuracy (no adversary). The authors should comment on the relation to the papers** :
> 1) https://aamas.csc.liv.ac.uk/Proceedings/aamas2014/aamas/p485.pdf
> 2) https://www.ncbi.nlm.nih.gov/pmc/articles/PMC3156487/
> 3) https://www.cs.waikato.ac.nz/~ml/publications/2002/bounds.pdf
> 4) https://ieeexplore.ieee.org/stamp/stamp.jsp?tp=&arnumber=7044723
>
> **and cite them, as it seems they study a similar question (possibly in the non-adversarial setting).** ”
>
> Thank you for providing these references. We will augment the related works section to provide a more detailed comparison.
>
> As you pointed out, apart from reference (1), all the provided references tackle the limits of deterministic ensembles in the absence of an adversary, which is completely unrelated to our problem setup. Reference (1) studies the performance of randomized ensemble classifiers in the context of adversarial reverse engineering. In this context, the defender trains a classifier $h$ from a concept class $H$ in order to distinguish between ‘good’ or ‘bad’ features $x$, and the adversary attempts to reverse engineer the classifier $h$ using queries. In contrast to our setting, where the defender is trying to learn a $C$-ary classifier $f$, and the adversary is allowed to perturb its inputs within some closed and bounded set $S$. Despite the different settings, Ref (1) derives a similar result to our Theorem 1 (Theorem 3.1 in (1)), where the optimal sampling probability for two classifiers is either uniform sampling or one of the two classifiers. Apart from this, there is no other connection. We will be sure to refer to (1) in the related works section.
>
> Finally, we hope that we have addressed your questions/concerns, and hope that you increase your score.

---

> > ### Comment · Reviewer_3zJ9 · 2022-11-16
> > **Updated paper?**
> >
> > I have read your rebuttal. I will update my review in the coming days. Will you upload a revised version of the paper during the rebuttal period?

---

> > > ### Author Response · Authors · 2022-11-16
> > > **Updated paper will be uploaded during the rebuttal period.**
> > >
> > > Most certainly, we will upload the revised paper addressing reviewer comments within the rebuttal period. Thanks for asking.

---

> ### Author Response · Authors · 2022-12-07
> **Reply to Reviewer Feedback**
>
> Thank you so much for updating your review and score! We appreciate the time and effort for understanding the contributions of our work.

---

### Official Review · Reviewer_5kPi · 2022-10-25

**Confidence:** 3
**Correctness:** 2
**Technical Novelty And Significance:** 3
**Empirical Novelty And Significance:** 3
**Recommendation:** 5

**Clarity, Quality, Novelty And Reproducibility:**

The paper is relatively clear, but could be more notationally clear in a few sections. The provided intuitions are nice and help tho clearly understand what the authors mean. From the details in the paper the method would be reproducible, but one would need access to the code base to reproduce the exact numbers in this paper as there are inevitably hyper-parameters that are not reported here.

**Details Of Ethics Concerns:**

None found.

**Strength And Weaknesses:**

Strengths:

The authors provide promising empirical evidence that an REC can be made from an ensemble of classifiers such that one gets performance gains over the naive ensembling approach that would be typically taken.

Weaknesses:

There are a couple of non-trivial drawbacks to this method that are not discussed. For the first contribution (having better computational complexity) the authors leave undiscussed the cost of this computational complexity. In particular, when using an REC one forfeits the uncertainty properties of an ensemble classifier (or Bayesian NN) which have been shown in many instances to help adversarial robustness and to aid in safety analysis. It feels like this is an important drawback: the computational gain here is not for free, you are sacrificing uncertainty. And this needs to be at the very least discussed if not empirically analyzed.

The theoretical results themselves are also simplistic, bordering on trivial. I do note here that I am not an expert on RECs, so perhaps reviewers with more experience in these models will find these results more interesting than I. In addition to being very simplistic the claims about some of the theorems are not correct. Most glaring to me is the statement that "there are no worst-case performance guarantees with deterministic ensembling, even if all the classifiers are robust." This statement is either unqualified and therefore misleading or simply  incorrect. A deterministic ensemble (or BNN) is a single classifier: a model average. Therefore, they _do_ fit into Theorem 2 as there is only one model under consideration in these cases and therefore only one $\eta$ (risk value) and therefore they do have the same guarantees. Thus the statement regarding Theorem 2 is incorrect. This also points to the fact that Theorem 2 is bordering on trivial because it can be said of any classifier and is not unique to RECs.

Finally, the results, while they do show a slight increase in performance, do not include any error bars. This seems a large omission given that the performance gains are ~1% in many cases and this is a randomized method so not reporting the variance is a red flag.

Minor note:  I find the placement of the related works to be a bit jarring. If it could be placed in before the problem statement that might help the flow of the paper. But this is stylistic.

**Summary Of The Paper:**

In this work, the authors analyze randomized ensembles through a theoretical perspective. A randomized ensemble classifier (REC) is an ensemble of classifiers where predictions are made using only one of the constituent models selected at random. The authors analyze this set up in an adversarial scenario and show a series of elementary facts about them including bounds on the worst-case risk of the REC and from these facts show how to set the probability that any one classifier is taken from the ensemble as the model from which a prediction is made. The primary motivations for this set up are claimed to be (1) greater computational complexity compared to ensembles (2) greater adversarial robustness compared to both ensembles and single models.

**Summary Of The Review:**

Overall, I found the theory presented to be simplistic bordering on trivial. The claims are also incorrect in places (see the example in weaknesses).  The experimental analysis does not really report error bars which is also a red flag. Overall I think the work needs some further developments to be impactful.

---

> ### Author Response · Authors · 2022-11-15
> **Response to Reviewer 5kPi (Part 1/2)**
>
> We appreciate your comments on our paper and have tried to address your concerns as best as we can.
>
> “**There are a couple of non-trivial drawbacks to this method that are not discussed. For the first contribution (having better computational complexity) the authors leave undiscussed the cost of this computational complexity. In particular, when using an REC one forfeits the uncertainty properties of an ensemble classifier (or Bayesian NN) which have been shown in many instances to help adversarial robustness and to aid in safety analysis. It feels like this is an important drawback: the computational gain here is not for free, you are sacrificing uncertainty. And this needs to be at the very least discussed if not empirically analyzed.** ”
>
> This is a good point. We will mention this limitation of RECs compared to BNNs. Please note though: we are not introducing RECs in this paper as an alternative to BNNs. RECs are already being studied by other researchers. Our paper sheds light on the limits of what RECs can achieve in the context of adversarial robustness.
>
> “**The theoretical results themselves are also simplistic, bordering on trivial. I do note here that I am not an expert on RECs, so perhaps reviewers with more experience in these models will find these results more interesting than I. In addition to being very simplistic the claims about some of the theorems are not correct. Most glaring to me is the statement that "there are no worst-case performance guarantees with deterministic ensembling, even if all the classifiers are robust." This statement is either unqualified and therefore misleading or simply incorrect. A deterministic ensemble (or BNN) is a single classifier: a model average. Therefore, they do fit into Theorem 2 as there is only one model under consideration in these cases and therefore only one $\eta$ (risk value) and therefore they do have the same guarantees. Thus the statement regarding Theorem 2 is incorrect. This also points to the fact that Theorem 2 is bordering on trivial because it can be said of any classifier and is not unique to RECs.** ”
>
> Please note we have provided a rigorous proof of correctness of Theorem 2 in Appendix A. Regarding your specific comments, we believe there is a misunderstanding about what Theorem 2 is saying, which we clarify below.
> Theorem 2 obtains the upper and lower bounds on the adversarial risk of a randomized ensemble of $M$ classifiers given a dataset $D$ and perturbation set $S$. Our statement that "there are no worst-case performance guarantees with deterministic ensembling, even if all the classifiers are robust” is a restatement of the fact that there is no equivalent of Theorem 2 for deterministic ensembles. In fact to prove this, we present an example in Appendix B.2 where a 2-classifier deterministic ensemble is constructed whose adversarial risk is strictly worse than those of its constituent classifiers.
> Your suggestion of taking the model average of a deterministic ensemble and using Theorem 2 leads to the trivial ($M=1$) but correct conclusion that the risk of a single classifier is equal to its own risk. It does not give any insight into the relationship between the individual classifiers’ risk and the risk of the deterministic ensemble. This is why Theorem 2 does not apply to deterministic ensembles.

---

> > ### Author Response · Authors · 2022-11-15
> > **Response to Reviewer 5kPi (Part 2/2)**
> >
> > “**Finally, the results, while they do show a slight increase in performance, do not include any error bars. This seems a large omission given that the performance gains are ~1% in many cases and this is a randomized method so not reporting the variance is a red flag.** ”
> >
> > There seems to be a misunderstanding here. In Tables 1 and 2 we report the EXACT empirical adversarial risk of randomized ensemble classifiers measured on the test sets. We can compute the exact risk due to the discrete nature of RECs, thus error bars are not required for this.
> >
> > However, we agree that reporting error bars to account for randomness of the training process itself due to SGD would be better. Please note that obtaining error bars for this, i.e., repeating the entire training run multiple times, is very prohibitive due to the long training times suffered by boosting frameworks in general. None of the past works such as BAT or MRBoost have shown error bars for this reason.
> > Finally, note that we are claiming that our method (BARRE) results in an ensemble that achieves similar adversarial risk as MRBoost but with much lower complexity. We do not claim better adversarial risk has been achieved. The point of our paper is to point out the limitations of RECs theoretically (this is the main contribution) and then suggest a training algorithm BARRE for RECs inspired by theory (this is a secondary contribution).
> >
> > “**Minor note: I find the placement of the related works to be a bit jarring. If it could be placed in before the problem statement that might help the flow of the paper. But this is stylistic.** ”
> >
> > We will be happy to accommodate your suggestion to place the related works before the problem statement.
> >
> > “**From the details in the paper the method would be reproducible, but one would need access to the code base to reproduce the exact numbers in this paper as there are inevitably hyper-parameters that are not reported here.** ”
> >
> > *All* the hyperparameters are reported in the appendix. The code has already been provided in the supplementary material.
> >
> > We hope that we have addressed all your concerns and cleared any misunderstandings of our work.

---

> > > ### Comment · Reviewer_5kPi · 2022-12-07
> > > **Thanks for your responses**
> > >
> > > I have read the responses by the authors to my review and their responses to other authors. They have done a good job addressing my concerns (with one minor exception below) and I have increased my score.
> > >
> > > For Theorem 2 I understand that their is no similar statement that can be made for deterministic ensembles, but BNNs and deterministic ensembles are typically used in a way that induces a classifier with M=1 as one marginalizes over the posterior (for BNNs) or uses model averaging (for deterministic ensembles) when making a prediction.

---

> > > > ### Author Response · Authors · 2022-12-07
> > > > **Reply to Reviewer Feedback**
> > > >
> > > > Thank you for updating your score. We really appreciate it.
> > > >
> > > > Regarding your remaining concern:
> > > >
> > > > You are correct that deterministic ensemble classifiers (and marginalizing over the posterior of BNNs) reduce to an M=1 deterministic classifier.
> > > > Our statement is slightly different: If you were given $M$ classifiers with their individual adversarial risks, then there is no guarantee (similar to Theorem 2) on the risk of the averaged classifier knowing **only** the individual risks.
> > > > Thus both your statement and ours are correct.

---

### Official Review · Reviewer_BAMT · 2022-10-31

**Confidence:** 4
**Clarity, Quality, Novelty And Reproducibility:** See comments above
**Correctness:** 4
**Technical Novelty And Significance:** 3
**Empirical Novelty And Significance:** 2
**Recommendation:** 3

**Strength And Weaknesses:**

Strengths:
  *  Randomized ensembling is an interesting defense strategy that is under-explored in the literature. While there are some works that provide algorithms for building robust RECs, they are easily broken by adaptive defenses. So there is a need for better randomized ensembling strategies. The paper takes a step towards solving this problem. The theoretical results provided in the paper on bounding the adversarial risk of RECs, are novel and interesting. However, it is not immediately clear how the insights gained from these theoretical results can be used to develop a good algorithm (more on this below).
  *  The paper is easy to read. The clarity and presentation is good. The proofs are elementary and easy to follow.

Weaknesses:
  *  Motivation:
      - The authors motivate the paper by saying that randomized ensembles can be computationally more efficient than deterministic ensembles. But there is a drawback of randomized ensembles that hasn't been brought up in the paper. In many application domains (e.g., healthcare), it is important to have deterministic predictions (otherwise it is hard to trust and understand these complex models). By moving to randomized ensembles, we lose this property. Can the authors provide some concrete use cases for randomized ensembles?
      -  Low compute resource devices also tend to have low memory. But RECs have high memory requirement ( same as deterministic ensembles). Given this, and the fact that deterministic ensembling techniques (like MRBoost) have better performance than BARRE, it is not entirely clear what the practical applications of RECs could be.


  * Empirical results:
     - the empirical results look weak. For example, in table 1, MRBoost has better performance than BARRE both in terms of robust and standard accuracy. This seems to be in contrast with the message one gets by reading the theory section in the paper. For example in page 6 (in paragraph titled "implications of upper bound"), it is mentioned that deterministic ensembles have much worse performance in the worst-case. Why is there this mismatch between theory and experiments?
         - under what circumstances are deterministic ensembles better than RECs and vice-versa?

 * BARRE:
    - the algorithm looks almost identical to MRBoost, except for line 12 in Algorithm 2. In BARRE, the weights for each component classifier in the ensemble are recomputed after every boosting iteration. Whereas in MRBoost, all the component classifiers are given equal weights. Given this, why can't we simply add step 12 to MRBoost and get a randomized classifier out of it? How would the resulting algorithm compare with BARRE?
    - What is the computational overhead of BARRE over MRBoost?  It looks like computing weights (line 12) can be expensive, especially for large M.
    - In the introduction the authors claim that BARRE is based on the theoretical results in the paper (page 2). But I don't see any connection between the two. In particular, I don't see how the theoretical results in sections 3.2, 3.3 are used to derive this algorithm.

 *  Theoretical Results:
    -  A number of insights on how to obtain robust RECs have been provided in the discussion after theoretical results. Can these insights be used to derive a better algorithm than BARRE?

 * Minor comments:
    -  a more detailed explanation on why K=5 in section 3.2 would be helpful to the readers
    -

**Summary Of The Paper:**

The paper considers the problem of adversarial robustness. For this problem the paper investigates the usefulness of randomized ensemble classifiers (REC) where one classifier is randomly selected from the ensemble during inference.  The main motivation behind considering RECs over deterministic ensembles is that the former has much smaller inference time than the latter.

The paper makes two main contributions. The first contribution, which is on the theory front, involves careful characterization of the adversarial risk of RECs. In particular, the authors obtain reasonably tight upper and lower bounds for the adversarial risk of RECs that depend on the adversarial risks of the component classifiers in the ensemble. Based on these bounds, the authors provide some useful insights on how to design RECs with good adversarial risk guarantees. The second contribution is to provide a boosting style algorithm (BARRE) for constructing robust RECs that can tolerate adversarial attacks. The algorithm is mostly inspired by a recently proposed robust boosting technique called MRBoost.

**Summary Of The Review:**

The clarity and presentation in the paper are good. The theoretical results are interesting and novel. But their usefulness is a little bit unclear. The empirical results look weak. Moreover, the proposed algorithm looks identical to MRBoost, except for a minor step which involves setting the weights of the component classifiers in the ensemble. Given this, I'm a little bit inclined towards rejecting the paper. But I'm happy to upgrade my score if the authors address my concerns.

---

> ### Author Response · Authors · 2022-11-15
> **Response to Reviewer BAMT (Part 1/3)**
>
> We appreciate your comments and suggestions. Our responses are provided below.
>
> "**The paper takes a step towards solving this problem. The theoretical results provided in the paper on bounding the adversarial risk of RECs, are novel and interesting. However, it is not immediately clear how the insights gained from these theoretical results can be used to develop a good algorithm**"
>
> We are glad you find our work novel and interesting. The connection between theory and the training algorithm can certainly be elaborated upon.
>
> "**The paper is easy to read. The clarity and presentation is good. The proofs are elementary and easy to follow.**"
>
> We did spend a lot of effort in simplifying the proofs and presentation. We are happy you noticed.
>
> "**The authors motivate the paper by saying that randomized ensembles can be computationally more efficient than deterministic ensembles. But there is a drawback of randomized ensembles that hasn't been brought up in the paper. In many application domains (e.g., healthcare), it is important to have deterministic predictions (otherwise it is hard to trust and understand these complex models). By moving to randomized ensembles, we lose this property. Can the authors provide some concrete use cases for randomized ensembles?**"
>
> We don’t quite understand your phrase “deterministic predictions”. All predictions made by deep nets are statistical in nature, whether they are generated by randomized or deterministic ensembles or single classifiers. Randomized ensembles therefore do not impose any special constraints on the space of potential applications. It can be employed in all applications where single classifiers or deterministic ensembles can be used.
>
> "**Low compute resource devices also tend to have low memory. But RECs have high memory requirement (same as deterministic ensembles). Given this, and the fact that deterministic ensembling techniques (like MRBoost) have better performance than BARRE, it is not entirely clear what the practical applications of RECs could be.**"
>
> Please note that RECs have the same memory requirements as deterministic (as we point out in Sec 4.1 page 9) and as you point out above, but with much smaller computational requirements. This reduction in number of floating-point operations per inference achieved by RECs (compared to deterministic ensembles) leads to direct reductions in both prediction latency and energy, both of which are crucial for always-on prediction applications where Edge devices are employed. Moreover, Table 1 and Figure 1 demonstrate that BARRE-trained RECs are able maintain its performance to within 0.5% of MRBoost-trained deterministic ensembles. This implies that RECs are overall better for any safety-critical edge application compared to deterministic ensembles.
>
> In any case, our main focus here is to develop a deep theoretical understanding of randomized ensembles’ adversarial robustness since as you point out in the ‘Strengths’ that this defense strategy is under-explored in the literature.

---

> > ### Author Response · Authors · 2022-11-15
> > **Response to Reviewer BAMT (Part 2/3)**
> >
> >
> > "**the empirical results look weak. For example, in table 1, MRBoost has better performance than BARRE both in terms of robust and standard accuracy. This seems to be in contrast with the message one gets by reading the theory section in the paper. For example in page 6 (in paragraph titled "implications of upper bound"), it is mentioned that deterministic ensembles have much worse performance in the worst-case. Why is there this mismatch between theory and experiments? under what circumstances are deterministic ensembles better than RECs and vice-versa?**"
> >
> > There seems to be a misunderstanding. Table 1 shows that BARRE achieves adversarial robustness to within 0.5% of MRBoost but with a much smaller (2X-4X) computational complexity (FLOPS). This is the main benefit of randomized ensembles trained via BARRE.
> > Furthermore, we don’t see any inconsistency between our theory and experiments. The statement you are referring to on page 6 appears in the paragraph is reproduced below:
> >
> > “Implications of upper bound: Intuitively, we expect that a randomized ensemble cannot be worse than the worst performing member (in this case $f_M$). A direct implication of this is that if all the members have similar robustness $\eta_i\approx\eta_j \forall i, j$, then randomized ensembling is guaranteed to either improve or achieve the same robustness. In contrast, deterministic ensemble methods that average logits (Zhang et al., 2022; Abernethy et al., 2021; Kariyappa & Qureshi, 2019) do not even satisfy this upper bound (see Appendix B.2). In other words, there are no worst-case performance guarantees with deterministic ensembling, even if all the classifiers are robust.”
> >
> > This statement is saying it is possible for a deterministic ensemble to have adversarial robustness that is worse than the least robust classifier in the ensemble (please see Appendix B.2 where we construct an example to illustrate this scenario). This is to highlight the fact that RECs do not suffer from this catastrophic scenario. It does not imply that all deterministic ensemble methods will. Thus, there is no inconsistency here.
> >
> > "**the algorithm looks almost identical to MRBoost, except for line 12 in Algorithm 2. In BARRE, the weights for each component classifier in the ensemble are recomputed after every boosting iteration. Whereas in MRBoost, all the component classifiers are given equal weights. Given this, why can't we simply add step 12 to MRBoost and get a randomized classifier out of it? How would the resulting algorithm compare with BARRE?**"
> >
> > Actually, MRBoost and BARRE are different in the following sense: 1) BARRE frequently updates the sampling probability via OSP (whose optimality is derived from our theoretical results) whereas MRBoost averages the classifier outputs, and 2) BARRE uses adaptive PGD to attack the randomized ensemble (line 8), whereas MRBoost uses PGD to attack the deterministic ensemble. These differences explain why MRBoost cannot be used to generate robust RECs.
> >
> > Specifically, our MobileNetV1 experiments from Table 2 highlight this difference, since the MRBoost trained REC with $M=5$ achieves worse performance than the SAME ensemble with $M=4$, that is MRBoost RECs  can disregard (assign zero probability) to the last classifier $f_5$. Simply adding OSP to MRBoost will not alleviate this issue, as it will disregard the last classifier during training.
> > In contrast, the BARRE trained MobileNetV1 REC assigns the same probability to all classifiers and achieves better performance than its MRBoost counterpart. This further explains why simply adding OSP to MRBoost does not work.
> >
> > "**What is the computational overhead of BARRE over MRBoost? It looks like computing weights (line 12) can be expensive, especially for large $M$**"
> >
> > The computational overhead of BARRE over MRBoost is minimal. The attack generation (line 8) routine has similar complexity to that of MRBoost, since both require computing gradients over M models. The OSP update step is applied intermittently every 10 epochs of training for each classifier (see App C.1 for full details). Furthermore, the OSP routine complexity is significantly reduced for $M\leq3$ using the results of Theorem 1 and Theorem 4 (App B.1) as we point out in Sec 4. Hence (over a complete run with $M=5$), we see a slight overhead in complexity (roughly 10%).

---

> > > ### Author Response · Authors · 2022-11-15
> > > **Response to Reviewer BAMT (Part 3/3)**
> > >
> > >
> > >
> > > “**In the introduction the authors claim that BARRE is based on the theoretical results in the paper (page 2). But I don't see any connection between the two. In particular, I don't see how the theoretical results in sections 3.2, 3.3 are used to derive this algorithm.**"
> > >
> > > Actually, Algorithms 1 and 2 are based on the theoretical results presented earlier. Page 8, Section 4, second paragraph clearly states that we use the Optimal Sampling Probability (OSP) algorithm (Algorithm 1) during BARRE training (Algorithm 2). The optimality of OSP (Theorem 3) is derived from the theoretical results. Combining the bounds in Theorem 2 and the optimality of OSP in Theorem 3 implies that an REC is guaranteed to achieve better or the same performance than the BEST performing member of the ensemble. This explains the choice of boosting for BARRE, since the first iteration reduces to standard AT, the most effective method to date for generating robust classifiers. Thus, BARRE trained RECs are guaranteed to perform better than AT.
> > >
> > > Furthermore, Theorem 1 (for the case of $M=2$) motivates the sequence of steps in BARRE as follows: Theorem 1 states that the optimal REC adversarial risk would be $\eta(\alpha^*) = 0.5(\eta_1+\eta_2-Pr(R_1))$ (assuming (8) is met), therefore it is equally important to minimize both eta’s and maximize $Pr(R_1)$. BARRE does this by initially adversarially training a robust classifier $f_1$ (minimizing $\eta_1$), then training $f_2$ (initialized from $f_1$ to minimizes $\eta_2$) on the adversarial examples of the REC of $f_1$ and $f_2$ (constructed with equiprobable sampling (due to Theorem 1)). Doing so increases $Pr(R_1)$ while maintaining $\eta_2$ as low as possible.
> > >
> > > Thus, Theorems 1, 2, and 3, lead to the formulation of BARRE. We can summarize these nuances in the paper.
> > >
> > > "**Theoretical Results: A number of insights on how to obtain robust RECs have been provided in the discussion after theoretical results. Can these insights be used to derive a better algorithm than BARRE?**"
> > >
> > > Please note that BARRE is the best algorithm currently for training randomized ensembles. BAT [Pinot et al., 2020] was previously the best but it was recently broken by both [Zhang et al., 2022; Dbouk & Shanbhag, 2022]. The theoretical results show what randomized ensembles are capable of and have already motivated BARRE. However, those results can certainly be used to further improve upon BARRE. We hope researchers in the community will take up this challenge alongside us.
> > >
> > > "**a more detailed explanation on why K=5 in section 3.2 would be helpful to the readers**"
> > >
> > > We can certainly provide a more detailed explanation on why $K=5$ in Section 3.2.Specifically, $K=5$ is obtained by simply enumerating the manifold ways in which two classifiers in a 2-classifier ensemble can make errors in the perturbation set S, e.g., Configuration 1 is when both classifiers can make errors but not simultaneously, i.e., with the same perturbation. Similarly, Configuration 5 is when both classifiers are correct for all perturbations in the $S$.
> > >
> > >
> > > Finally, we hope that we have addressed your concerns and cleared all misunderstanding. We hope that you reconsider your score.

---

> ### Author Response · Authors · 2022-12-07
> **Waiting for Reviewer Feedback**
>
> Dear Reviewer BAMT,
>
> Thank you again for your comments and suggestions. We have provided extensive responses to your feedback. As the discussion period soon comes to an end, we hope you will read our responses, provide additional feedback if necessary, and revise your score as you indicated in your review. Thanks again.

---

### Author Response · Authors · 2022-11-16
**Uploaded Revised Manuscript**

Dear Reviewers,

We have uploaded a revised version of the manuscript. Changes are marked in red. Below is a summarized list of changes:

- moved the related works section before the preliminary section (Section 2)
- added brief discussion of Bayesian Neural Nets in related works (Section 2)
- briefly mention the difficulty of maximizing $Pr(R_1)$ in practice (Section 4.2)
- mention the connection between [Vorobeychik & Li (2014)] and Theorem 1 (Section 4.2)
- added definition of FLOPs (Section 5)
- new section on the theoretical rationale behind BARRE (Appendix A)

We thank the reviewers again for their valuable feedback, and hope that the revised manuscript clarifies most of your concerns/questions.

---

### Decision · Program_Chairs · 2023-01-20

**Decision:**

Reject

**Justification For Why Not Higher Score:**

There are still some open points and split opinions among reviewers. I personally think that doubts and some other issues would be fixed if this was a longer reviewing process of a journal, but this is a guesstimate. Given the information that I have at hand and the highly competitive nature of ICLR in terms of space for contributed work during the conference, this submission remains uncertain to me in order to give a more positive recommendation.

**Justification For Why Not Lower Score:**

N/A

**Metareview: Summary, Strengths And Weaknesses:**

Interesting results on the quality of randomised ensemble classifiers (RECs). The paper has some nice observations about the piece-wise linearity of the risk, which leads to a few theoretical results and an algorithm. Under a certain viewpoint, results are not so sophisticated, but they may have good use. Impact may well depend on whether RECs are appropriate for a particular task, and this seems to be another discussion point too. There have been a few points of attention, and authors have tried to clarify them in the discussions. The paper is arguably heavy (even if the underlying results might not be so mathematically heavy) and it might not be trivial to follow the importance of the theoretical results without resorting to the appendices. Unfortunately this requires extra time that some might not have in a short and super-busy reviewing process. Initially reviewers were mostly negative, and during the reviewing process a couple of reviewers improved their scores and viewpoint about the submission. However, this has not resolved the split opinions about the submission. It is apparent from the discussion that some points have not been fully resolved.